**Measurement report: Spatial variations in ionic chemistry and water stable isotopes in the snowpack on glaciers across Svalbard during the 2015-2016 snow accumulation season**

Elena Barbaro[1,2], Krystyna Koziol[3], Mats P. Björkman[4], Carmen P. Vega[5], Christian Zdanowicz[6], Tonu Martma[7], Jean-Charles Gallet[8], Daniel Kępski[9], Catherine Larose[10], Bartłomiej Luks[9], Florian Tolle[11], Thomas V. Schuler[12,13], Aleksander Uszczyk[14] and Andrea Spolaor[1,2]*

[1]Institute of Polar Sciences, ISP-CNR, Via Torino 155, 30170 Venice Mestre, Italy

[2]Department of Environmental Sciences, Informatics and Statistics, Ca' Foscari University of Venice, Via Torino 155, 30172, Venice, Italy.

[3]Department of Analytical Chemistry, Chemical Faculty, Gdansk University of Technology, G. Narutowicza 11/12, 80-233 Gdańsk, Poland.

[4]Department of Earth Sciences, University of Gothenburg, Box 460, SE-40530 Gothenburg, Sweden.

[5]Dirección Meteorológica de Chile, Dirección General de Aeronáutica Civil, Portales 3450, Santiago, Chile. *Previously at:* Department of Earth Sciences, Uppsala University, Villavägen 16, Uppsala, Sweden.

[6]Department of Earth Sciences, Uppsala University, Villavägen 16, SE-76236, Uppsala, Sweden.

[7]Department of Geology, Tallinn University of Technology, Ehitajate tee 5, 19086 Tallinn, Estonia

[8]Norwegian Polar Institute, Tromsø, No-9296, Norway

[9]Institute of Geophysics, Polish Academy of Sciences, Księcia Janusza 64, 01-452 Warsaw, Poland

[10]Environmental MicrobialGenomics, Laboratoire Ampère, CNRS, University of Lyon, France

[11]Université de Franche-Comté, Besancon, FEMTO-ST, UMR 6174 CNRS

[12]Departement of Geosciences, University of Oslo, Oslo, Norway

[13]Arctic Geophysics, University Centre in Svalbard, UNIS, Longyearbyen, Svalbard, Norway

[14]University of Silesia in Katowice, Faculty of Natural Sciences, Będzińska 60, 41-200 Sosnowiec, Poland

*Corresponding author

Andrea Spolaor (andrea.spolaor@cnr.it)

**Keywords**

Snow, Svalbard, Arctic, inorganic ions, water isotopes

**Abstract**

The Svalbard archipelago, located at the Arctic sea ice edge between 74° and 81° N, is ~60% covered by glaciers. The region experiences rapid variations in atmospheric flow during the snow season (from late September to May) and can be affected by air advected both from lower and higher latitudes, which likely impact the chemical composition of snowfall. While long-term changes in Svalbard snow chemistry have been documented in ice cores drilled from two high-elevation glaciers, the spatial variability of the snowpack composition across Svalbard is comparatively poorly understood. Here, we report the results of the most comprehensive seasonal snow chemistry survey to date, carried out in April 2016 across 22 sites on 7 glaciers across the archipelago. At each glacier, three snowpits were sampled along the altitudinal profiles and the collected samples were analysed for major ions ($Ca^{2+}$, $K^+$, $Na^+$, $Mg^{2+}$, $NH_4^+$, $SO_4^{2-}$, $Br^-$, $Cl^-$ and $NO_3^-$) and stable water isotopes ($\delta^{18}O$, $\delta^2H$). The main aims were to investigate the natural and anthropogenic processes influencing the snowpack and to better understand the influence of atmospheric aerosol transport and deposition patterns on the snow chemical composition. The snow deposited in the southern region of Svalbard is characterized by the highest total ionic loads, mainly attributed to sea salt particles. Both $NO_3^-$ and $NH_4^+$ in the seasonal snowpack reflect secondary aerosol formation and post-depositional changes, resulting in very different spatial deposition patterns: $NO_3^-$ has its highest loading in the northwestern Spitsbergen, and $NH_4^+$ in the southwest. The $Br^-$ enrichment in snow is highest in northeastern glacier sites closest to areas of extensive sea ice coverage. Spatial correlation patterns between $Na^+$ and $\delta^{18}O$ suggest that the influence of long-range transport of aerosols on snow chemistry is proportionally greater above 600-700 m a.s.l.

## 1. Introduction

Svalbard is a region of the Arctic experiencing rapid climate change. The mean warming rate is +1.35 K per decade, much faster than the global average (Isaksen et al., 2016; Maturilli et al., 2013; Nordli et al., 2014). This archipelago is located at the southern edge of the perennial Arctic sea ice in the North Atlantic Ocean, and is characterized by a maritime climate with large, rapid temperature variations during winter (Brage et al., 2014). South-westerly inflow of mild oceanic air, associated with a low-pressure system east of Iceland, often brings relatively warm and moist air in the winter months, while Arctic air intrusions from the north-east, driven by a high-pressure system over Greenland, result in much colder temperatures (Rinke et al., 2017). In addition to these synoptic fluctuations, intense autumn or winter cyclonic storms associated with anomalous warming events sometimes occur, transporting both heat and moisture from lower latitudes to Svalbard (Rinke et al., 2017).

The aforementioned meteorological conditions also favor long-range transport of aerosols to the archipelago, including pollutants from continental sources. Depending on the predominant air flow pattern at the time of snowfall, the archipelago may experience regionally different amounts of both snow accumulation (Eneroth et al., 2003; Forland et al., 2011) and chemical loads, the latter reflecting

contrasting mixtures of aerosols, varying by source area (Aas et al., 2016; Forsström et al., 2009; Möller and Kohler, 2018). These regional differences are also associated with contrasts in sea ice cover. While all Svalbard coasts are usually ice-free in summer, sea ice can form and cover large parts of the ocean surface in the eastern and northern parts of the archipelago. Contrastingly, the southern and western parts often remain ice-free (Dahlke et al., 2020), and therefore tend to experience greater snowfall owing to the proximity of open water. In addition, the West Spitsbergen Current, a branch of the Atlantic Meridional Overturning Circulation (AMOC) that flows to the west of the archipelago, causes markedly different regional climatic conditions between its eastern and western parts (van Pelt et al., 2019): the west exhibits higher temperatures and precipitation, while the east is less humid and cooler, and has also experienced a stronger warming trend since 1957.

The seasonal snowpack contains a complex mixture of impurities that are either scavenged from the atmosphere during snowfall or directly received through dry deposition (Kuhn, 2001). On land, the majority of impurities found in seasonal snow are usually eluted during summer melt, influencing terrestrial and aquatic systems (Björkman et al., 2014; Brimblecombe et al., 1987). However, in the accumulation area of Arctic glaciers and ice caps, impurities can be retained within or below the seasonal snow layer (Björkman et al., 2014; Pohjola et al., 2002; Vega et al., 2015b). For this reason, chemical impurities such as major ions ($Ca^{2+}$, $K^+$, $Na^+$, $Mg^{2+}$, $NH_4^+$, $SO4^{2-}$, $Br^-$, $Cl^-$ and $NO_3^-$) in ice cores have been widely used to study the past trends of atmospheric and climatic conditions (Barbante et al., 2017; Isaksson et al., 2003; Thompson et al., 2002; Wolff et al., 2010). Previous studies in Svalbard (Isaksson et al., 2001; Matoba et al., 2002; Nawrot et al., 2016; Semb et al., 1984) have shown that the chemistry of the seasonal snowpack is dominated by sea salt ions (Hodgkins and Tranter, 1998). However, the region is also a sink for atmospheric contaminants brought in by long-range transport (Vecchiato et al., 2018). Investigations of precipitation and snow cover chemistry have predominantly focused on the central and western parts of the archipelago (Kühnel et al., 2011; Nawrot et al., 2016; Vega et al., 2015a; Virkkunen et al., 2007), due to the accessibility of research facilities in these sectors.

In general, stable water isotope measurements in different components of the water cycle are available in isotope databases, maintained and updated by the International Atomic Energy Agency (IAEA), but also by national or international organizations (West et al., 2010). Moreover, in Svalbard, stable water isotope investigations are performed in ice cores and surface snowpit samples because $\delta^{18}O$ and $\delta^2H$ are still the most common tools for finding the depth-time relationship in ice cores (Pohjola et al., 2017; Punning et al., 1986). The preservation of un-interrupted annual isotope cycles varies depending on the site: in sites such as central Greenland annual isotope cycles are well preserved, while in sites with high intra-seasonal variations or with different pre- and post-depositional processes the annual layers can be difficult to distinguish (Pohjola et al., 2002)). To investigate to what degree $\delta^{18}O$ in snow changes after accumulation, Igarashi et al. (2001) carried out the observation of the precipitation at Ny-Ålesund. These authors concluded that the fluctuation of $\delta^{18}O$ could not be explained by changes in surface air temperature only,

but that the characteristics of the air masses also influenced the isotope signature of the precipitation
(lgarashi et al., 2001). At the moment, there is a lack of data regarding the stable water isotope composition
of surface snow from Svalbard and this survey is a substantial contribution to fill that gap.
In the present study, the concentration, mass loading, spatial and altitudinal distribution of major ion
species ($Ca^{2+}$, $K^+$, $Na^+$, $Mg^{2+}$, $NH_4^+$, $SO_4^{2-}$, $Br^-$, $Cl^-$ and $NO_3^-$) in snow, together with its stable oxygen and
hydrogen isotope composition ($\delta^{18}O$ and $\delta^2H$), were evaluated in the late winter snowpack at 22 glacier
sites across Svalbard. Stable isotope ratios were used as supporting data to define the accumulation
seasonality in the snowpack, and to identify the moisture sources that feed snowfall, thereby providing
clues to the predominant air transport pathways to the snowpit sites (Gat et al., 2001).This study was part
of the larger Community Coordinated Snow Study in Svalbard (C2S3) project and is the most
comprehensive survey of seasonal snow chemistry in Svalbard to date. The snowpack survey, which was
carried out by coordinated teams using a common sampling protocol (Gallet et al., 2018), aimed to map
and characterize regional differences in the chemical composition and impurity load of the winter
snowpack. We further interpret the observed differences in chemical loading in relation to meteorological
and other environmental factors.
Thereby, we aim to identify the conditions controlling the chemistry of Svalbard snow that are susceptible
to the variable climate warming impact across the region.

**2. Methods**
*2.1 Sampling location and strategy*
During April 2016, the seasonal snowpack was sampled at 22 sites on seven glaciers across Svalbard (Table
1 and Figure 1). The glaciers are of different sizes and hypsometry. Wind fields for each glacier are not
available. Indeed the wind direction can change in concomitance with cyclonic events that could occur
during the season and can act differently for each glacier.
The first glacier considered in this study is Austfonna (AF), located on Nordaustlandet, the second largest
island in Svalbard, with approximately 80% of its area covered by ice. AF is the largest ice cap in Svalbard
with a geographic area of 8357 $km^2$ and has one main central dome of up to 600 m ice thickness feeding
several drainage basins (Dallmann, 2015; Schuler et al., 2020).
The other six glaciers investigated here are located on the Spitsbergen Island. On the northwestern
Spitsbergen, we studied three glaciers near Ny-Ålesund: Austre Lovénbreen (ALB), Kongsvegen (KVG),
and Holtedahlfonna (HDF). ALB is a small land-based valley glacier, 4 km long from south to north along
the Brøgger Peninsula. The glacier area was 4.48 $km^2$ in 2013 and its elevation ranges from 50 to 550 m
a.s.l. The total catchment area spreads over 10.577 $km^2$, taking into account an outlet where the main
stream crosses a compact calcareous outcrop 400 m upstream from the coastline (Marlin et al., 2017). KVG
is a northwest-flowing grounded glacier located about 20 km east of Ny-Ålesund (Melvold & Hagen 1998),
with an average ice thickness of 190 m and a maximum 450m (Lindbäck et al., 2018). It has a total length
of ca. 24 km with an average 3.5 km width, has a maximum elevation of 800 m a.s.l. and flows from south-
east to north-west (Spolaor et al., 2017). HDF glacier is the largest ice field (c.a. 300 km$^2$) on the
northwestern Spitsbergen Island, about 40 km from the Ny-Ålesund station. It is distributed over an
elevation range of 0–1241 m a.s.l (Beaudon et al., 2017; Nuth et al., 2017).
Lomonosovfonna (LF) is one of the highest ice fields on Spitsbergen and it is located on the central part
of the island. The summit lies at 1250 m a.s.l. and has a pronounced cupola shape with an approximate
radius of 500 m. The total accumulation area of the entire LF ice system was about 600 km$^2$ at the beginning
of the 21$^{st}$ century (Isaksson et al., 2001). Even though this is the highest point in our survey, the air
temperature can pass above zero during the summer resulting, although not significant, in the relocation of
ions (Pohjola et al., 2002; Vega et al., 2016). In southern Spitsbergen, two different glaciers were
investigated, Hansbreen (HB) and Werenskiöldbreen (WB), close to the Hornsund station. The HB is a
medium-sized (56 km$^2$) tidewater glacier located in the southern part of Wedel Jarlsberg Land. The glacier
is ~16 km long, and its elevation extends up to 550 m a.s.l. The WSB glacier has an area of 27 km$^2$, is a
land-terminating valley glacier to the west of HB, and ranges in elevation from 50 to 600 m a.s.l.(Schuler
et al., 2020).
Each glacier was sampled in the ablation zone, close to the equilibrium line altitude (ELA), and in the
accumulation zone (Table 1). The ELA is the elevation at which the surface mass balance is zero, i.e.,
where the accumulation of snow is exactly balanced by ablation over a period of a year (Cogley et al.,
2011). Although the exact elevation ranges of these zones (accumulation, ablation, and ELA) differ for
each glacier, they share enough glaciological similarities to support intersite comparisons. All snow pits
have been collected from the glacier central line in order to minimise the side accumulation effect due to
orography.

*2.2. Sampling procedure*
Snowpit sampling was performed using a common protocol (Gallet et al., 2018) with pre-cleaned
equipment (i.e., tubes, plastic scrapers, and plastic shovels cleaned with ultrapure water) and protective
clothing (powder-free plastic gloves, clean suits and face masks). This protocol allowed sampling and field
data collection in a consistent manner, obtaining comparable datasets from different research sites.
Samples for ionic chemistry were taken from each discrete snow pit layer, according to the visible
stratigraphy, and were directly filled into pre-cleaned 50 mL polypropylene "Falcon" centrifuge tubes. This
type of sampling facilitates linking a snow layer (and its properties) to a specific weather event (i.e.,
precipitation or surface melt). Moreover, sampling by discrete layers makes it possible to correlate the
intervals of snow accumulation between separate snow pits at different altitudes, as reported in this paper
when we compare three different areas of the same glacier (ablation, ELA and accumulation). It is also
more accurate for chemical load calculations where ice layers occur in snow pits.
Samples for the isotopic composition of water were collected at a 5-cm resolution for sites in the Ny-
Ålesund area and at a 10-cm or stratigraphic layer resolution for other sites. All sampling was conducted
at a safe distance and upwind from potential local pollution sources, such as the snowmobiles used for
transport by the sampling team.

***2.3 Major ion analyses***
Samples from glaciers in the Hornsund area (HB, WB) were analysed at the Polish Polar Station Hornsund
(Institute of Geophysics, Polish Academy of Sciences), while samples from glaciers near Ny-Ålesund
(KVG, ALB, HDF) were shipped frozen to the Institute of Polar Sciences (ISP-CNR) in Venice (Italy).
Snow sampled in central Spitsbergen (AF, LF) was shipped frozen to the Department of Earth Sciences at
Uppsala University (Sweden). Due to a temporary equipment malfunction in Uppsala, only cations could
be analysed there, and the refrozen samples were forwarded to ISP-CNR for anion analysis. All samples
and standards were handled and prepared under clean room conditions, wearing powder-free gloves. In all
labs except at the Polish Polar Station Hornsund, laminar flow hoods (class 100) were used. Samples were
melted immediately before analysis.

*2.3.1. Hornsund*
Samples were filtered through 0.45 µm mixed cellulose esters membranes (Merck Millipore S-pak®) prior
to analysis. Ion concentrations were determined on a Metrohm 761 Compact IC ion chromatograph
equipped with an autosampler (Metrohm, Herisau, Switzerland), with isocratic flow of 0.69 mL min$^{-1}$, and
chemical suppression for anions (column Metrosep A Supp S + Metrosep A Supp 4/5 Guard 4.0, eluent:
NaHCO$_3$ 1.0 mmol L$^{-1}$ + Na$_2$CO$_3$ 3.2 mmol L$^{-1}$). Cations were determined without suppression (column
Metrosep C4 + Metrosep C4 Guard; mobile phase: HNO$_3$ 1.7 mmol L$^{-1}$ + 2,6-pyridinecarboxylic acid
[dipicolinic acid, DPA] 0.7 mmol L$^{-1}$ at a flow rate of 0.9 mL min$^{-1}$). Cation samples were acidified with 2
µL of 2mM HNO$_3$ per 10 mL sample prior to analysis. The injection volume was 20 µL in the anion system
and 100 µL in the cation system. Nitric acid solutions were prepared from POCH S.A. (Poland)
concentrated weighed amounts, while sodium carbonate and hydrogen carbonate as well as DPA were
dissolved from the solid phase (Merck Millipore).

*2.3.2 Uppsala*
Samples were filtered using 0.22 µm polyethersulfone membranes (Minisart®, Sartorius) and cation
determination was performed using a Metrohm ProfIC850 ion chromatograph (Metrohm, Herisau,
Switzerland), equipped with an autosampler and a Metrosep C4 column. The mobile phase of 0.02 M DPA
and 0.1 M HNO$_3$ was run in isocratic flow of 0.7 mL min$^{-1}$. Very low detection limits ($\leq$ 0.006 mg $^{-1}$L)
were achieved thanks to the sample injection volume of 500 µL.

*2.3.3. Venice*
Anion determination was performed using a Dionex™ ICS-5000 ion chromatograph (ThermoScientific™,
Waltham, US) equipped with an anionic exchange column (Dionex IonPac AS 11, 2 × 250 mm) and a
guard column (Dionex IonPac AG11 2 × 50 mm). Sodium hydroxide (NaOH), used as a mobile phase, was
produced by an eluent generator (Dionex ICS 5000EG, Thermo Scientific). The gradient with a 0.25 mL
$min^{-1}$ flow rate was 0 min, 0.5 mM; 0–3.5 min, gradient from 0.5 to 5 mM; 3.5–5 min, gradient from 5 to
10 mM; 5–25 min, gradient from 10 to 38 mM; 25–30 min, column cleaning with 38 mM; 30–35 min;
equilibration at 0.5 mM. The injection volume was 100 μL. The IC was coupled to a single quadrupole
mass spectrometer (MSQ Plus™, Thermo Scientific™) with an electrospray source (ESI) that operated in
negative mode. All other details are reported by Barbaro et al. (2017).
To determine the cations, a capillary ion chromatograph (Thermo Scientific Dionex ICS-5000) equipped
with a capillary cation exchange column (DionexIonPac CS19-4μm, 0.4 × 250 mm) and a guard column
(Dionex IonPac CG19-4μm, 0.4 × 50 mm) coupled to a conductivity detector was used. Methanesulfonic
acid, produced by an eluent generator (Dionex ICS 5000EG, Thermo Scientific), was applied as mobile
phase. The gradient was 0 − 17.3 min: 1.5 mM; 17.3 − 21.9 min: from 1.5 to 11 mM; 21.9–30 min:
equilibration at 1.5 mM. The injection volume was 0.4 μL and the flow rate was 13 μL $min^{-1}$.

*2.3.4. Instrumental performance of each laboratory*
For all laboratories, calibrations for ions were evaluated using analytical standards (Merck/Sigma Aldrich).
The calibrations in each lab delivered different linear ranges for each ion due to the different methods used
(Table S1). Good linearity was demonstrated in each lab and all calibration curves had $R^2 > 0.99$. Samples
that had ion concentrations beyond the calibration range were diluted with ultrapure water before re-
analysis. Analytical blanks of ultrapure water (>18 MΩ cm) were included in the analysis at all three labs.
The method detection limit (MDL) was set to three times the standard deviation of the blank values (Table
S1). For $Na^+$, $Mg^{2+}$, $Cl^-$ and $SO_4^{2-}$, values < MDL occurred in less than 10% of cases, and for $Ca^{2+}$ and $NO_3^-$
the < MDL concentrations were noted in 12% and 17% of all cases, respectively. However, $K^+$ and $Br^-$
were detected only in 53% and 46% of all samples, respectively, while $NH_4^+$ concentration exceeded the
MDL only in 36% of all measurements. For the calculation of the bulk ionic loading in snowpits, values <
MDL were assumed to be equal to half the MDL.

Accuracy and precision are important parameters to be evaluated during method validation. Checks for
accuracy were made using certified multi-element standard solutions for anions ($F^-$, $Cl^-$, $Br^-$, $NO_3^-$, $SO_4^{2-}$,
n° 89886-50ML-F, Sigma Aldrich) and cations ($Na^+$, $K^+$, $Mg^{2+}$, $Ca^{2+}$, n° 89316-50ML-F, Sigma Aldrich),
at the concentration of 10 mg $L^{-1}$ ± 0.2%. Accuracy is expressed as a relative error calculated as
(Q−T)/T×100, where Q is the determined value and T is the "true" value. The accuracy for each ion in all
labs was always <±10%, except for $Mg^{2+}$ measurements at the Hornsund laboratory. The analytical
precision was quantified as the relative standard deviation (RSD) for replicates (n > 3) of standard solutions
and was always < 10% for each ion (Table S1).

**2.4. Stable water isotopes**

The determination of stable isotope ratios of O and H was performed at Tallinn University of Technology (Estonia). The isotopic ratios were determined by laser spectroscopy, using a Picarro model L2120-i water isotope analyzer (Picarro Inc., Sunnyvale, USA), which allows for the simultaneous determination of $^{18}O/^{16}O$ and $^2H/^1H$ in $H_2O$ with a high-precision AO211 vaporizer. Results are reported in the standard delta notation as $\delta^{18}O$ and $\delta^2H$ relative to Vienna Standard Mean Ocean Water (VSMOW). Reproducibility was ±0.1‰ for $\delta^{18}O$ and ±1‰ for $\delta^2H$, respectively. 7 injections were carried out for each sample, but only the last 4 injections (4 to 7) were used for calculations to minimize the memory effect. Laboratory standards TLN-A2 (-10.15; -77.5) and TLN-B2 (-21.95; -162.5) were regularly calibrated against international V-SMOW, GNIP and V-SLAP standards. Standards (TLN-A2, TLN-B, and TLN-D4) were measured at the beginning, in the middle, and at the end of each set of measurements (54 bottles). Additionally, every 7 samples, the laboratory standard TLN-D4 (-17.5; -133.0) was measured and used for drift correction if needed.

## 3. Results

### 3.1 Spatial distribution of ionic species

To investigate differences in snowpack composition across all glaciers, we compared the total mass of ions that accumulated in snow at the different sampling sites. On average, the snow cover season on Svalbard lasts from early September to early May, but snow may also fall in summer months at high elevations. The snow pits in this study were sampled in early to late April 2016 and therefore might not contain the full annual ionic burden, since deposition can still occur before the beginning of the snow melt season. Therefore, we report these data as ionic loads (mg m$^{-2}$) rather than annual fluxes. In each snow pit, the total ionic load was calculated as the cumulative sum of the ionic loads in each discrete layer, i.e., ionic concentrations multiplied by the snow water equivalent of the layer (Table 2). On the other hand, to evaluate the transport processes of chemical species from other regions to Svalbard, we evaluate the volume-weighted mean concentrations of major ions in each snow pit. These values are calculated as the total ionic load of each snow pit divided by its total SWE (snow water equivalent) (Table 3). The snowpack chemical characteristics were then compared between glacier zones (ablation zone, ELA, and accumulation zone; Figure 1).

Snow pit samples collected in the Hornsund area (HB and WB, southern Spitsbergen) show a markedly higher total load for all major ions (Figures 1 and 2) than at all other sites. The samples collected in the accumulation zones of WB and HB have total ionic loads of 8161 and 8023 mg m$^{-2}$, respectively, four times higher than those collected in the accumulation zone of KVG (2861 mg m$^{-2}$), AF (2607 mg m$^{-2}$) and ALB (1934 mg m$^{-2}$) and 16 times higher than those sampled at LF (639 mg m$^{-2}$) and HDF (583 mg m$^{-2}$). Similar differences are observed for the snowpits collected at lower altitudes (Figure 2).

289 In the accumulation zone of all glaciers (Figure 3), $Na^+$ and $Cl^-$ are generally the most abundant ionic

290 species, with percentages ranging from 29% (HDF) to 36% (AF) for $Na^+$, and from 34% (LF) to 48% (HB

291 and WB) for $Cl^-$, respectively. The snowpack on Hornsund glaciers (HB, WB) has higher $Cl^-$ percentages

292 (48–49%) compared to that of other glaciers (34–39%), while conversely the $SO_4^{2-}$ percentage is lower

293 there (9%) than on other glaciers (11–23%). The ionic loads are generally highest in the accumulation zone

294 of glaciers and lowest in the ablation zone (Figure 2), mostly due to the lower snow accumulation. This

295 pattern holds true for $Na^+$, $Cl^-$, $NH_4^+$, $K^+$, $Ca^{2+}$, and $Mg^{2+}$ at most glacier sites, except in the Hornsund area.

296 The load of $Br^-$ is similar on glaciers of the Ny-Ålesund sector (ALB, HDF, KVG) and on LF, but is higher

297 in AF and Hornsund glaciers (HB, WB; Figure 2). The load of $NO_3^-$ is similar for all glaciers, except for

298 LF, where very low loads are found. Unlike total $SO_4^{2-}$, the non-sea-salt fraction of sulphate (nss-$SO_4^{2-}$),

299 calculated using a seawater $SO_4^{2-}$:$Na^+$ mass ratio of 0.252 (Millero et al., 2008), shows lower loads on

300 Hornsund glaciers (15–107 mg m$^{-2}$) when compared to glaciers in other parts of the archipelago (Figure 1,

301 Table 2). The nss-$SO_4^{2-}$ loads vary between 22–131 mg m$^{-2}$ at HDF and LF, 75–266 mg m$^{-2}$ at KVG and

302 ALB, and 153–206 mg m$^{-2}$ at AF.

303

304 *3.2 Stable water isotopes ($\delta^{18}O$ and $\delta^2H$)*

305 Our results provide the first picture of spatial variations in the mean stable water isotope composition of

306 the seasonal snowpack across Svalbard (Table 3, Figure 4). The SWE-weighted mean $\delta^{18}O$ and $\delta^2H$ are

307 calculated using the formula SWE-$\delta = \sum(\delta_i \times SWE_i)/SWE_t$ where $\delta_i$ are the $\delta$ values of each layer, $SWE_i$

308 are the SWE of each layer and $SWE_t$ is the SWE of the entire snowpit. These SWE-weighted mean values

309 decrease significantly from south to north (Spearman rank correlation $\rho$ with latitude is -0.69 and -0.65 for

310 $\delta^{18}O$ and $\delta^2H$, with $p < 0.001$ and $p < 0.01$, respectively). The isotopically heaviest snow (least negative $\delta$

311 values) occurs on glaciers of the Hornsund area ($\delta^{18}O$: -11.25 to -9.54 ‰; $\delta^2H$: -77.62 to -63.64 ‰), and

312 the isotopically lightest (most negative $\delta$ values) in AF ($\delta^{18}O$: -16.00 to -13.89 ‰; $\delta^2H$: -111.15 to -96.89

313 ‰). Glacier sites in NW Spitsbergen (KVG, ALB, and HDF) and on LF have mean $\delta^{18}O$ and $\delta^2H$ values

314 that fall within these ranges. On KVG, ALB, HDF and LF, $\delta^{18}O$ and $\delta^2H$ values in snow decrease

315 monotonically (becoming gradually more negative) with increasing elevation. On the other hand, on AF,

316 WB, and HB there is no statistical difference between the mean $\delta^{18}O$ and $\delta^2H$ values of all snow pits (Figure

317 4). A general significant anticorrelation with altitude is found for SWE-weighted mean $\delta^2H$ ($\rho = -0.63$, $p$

318 $< 0.01$), and $\delta^{18}O$ ($\rho = -0.65$, $p < 0.01$).

319

320 **4. Discussion**

321

322 There have been few published studies on recent seasonal snow or firn chemistry in Svalbard, hence

323 comparisons of our data with these earlier results are limited to a few sites. Virkkunen et al. (2007) and

324 Vega et al. (2015a) quantified the annual chemical loads of $Na^+$, $Ca^{2+}$, $NO_3^-$ and nss-$SO_4^{2-}$ at

325 Lomonosovfonna summit (LF3) from 2002 to 2011 using snow and firn cores. In Table 4, we also report

the unpublished data of samples collected in 2009-2011 by C. Vega, obtained using the methods outlined
in Section 2.2. Our study extends these data to 2016. The range of annual ionic loads at LF3 over the 15-
year period is wide, and no clear temporal trend can be identified (Table 4). At Holtedahlfonna summit
(HDF3), firn core measurements by Spolaor et al. (2013) found a mean $Na^+$ concentration of $110 \pm 73$ ng
$g^{-1}$ over the period 2003-2012, while the mean concentration in the April 2016 snowpack (this study) was
191 ng $g^{-1}$, hence within the range reported in earlier years.

### 4.1 The main ion sources in the seasonal snow of Svalbard

The composition of the Svalbard seasonal snowpack sampled during the C2S3 project clearly indicates
that the ocean is the main source of ions in snow, as was shown by Hodgkins and Tranter (1998). At all
sites, the dominant ions are $Na^+$, $Cl^-$, and $SO_4^{2-}$, with comparatively minor amounts of $K^+$, $Ca^{2+}$, and $Mg^{2+}$
(Figure 3). To help clarify the possible sources and modes of deposition of ions in snow, we computed
Spearman rank correlations between total ionic loads ($\rho_{load}$), as well as between volume-weighted mean
ionic concentrations ($\rho_{conc}$), across all snowpits ($n = 22$; Table 5). The chemical species that are
predominantly wet-deposited, sharing common sources and not undergoing significant composition
changes in transport should exhibit similar concentration patterns (high $\rho_{conc}$) (Schüpbach et al., 2018). The
concentrations of $Mg^{2+}$, $K^+$, and $Ca^{2+}$ are all positively correlated with those of $Na^+$ and $Cl^-$, indicating a
common sea spray source. Moreover, this input is the single significant source of $K^+$ and $Mg^{2+}$, as indicated
by near-zero calculated values of nss-$K^+$ and nss-$Mg^{2+}$ in the sampled snowpits (Table 3). The $\rho_{load}$
correlations are very similar for these ionic species, which points to both wet and dry deposition being a
significant mechanism in their accumulation in the snowpack.

The concentrations of $Mg^{2+}$ are positively and significantly correlated with both $Ca^{2+}$ and nss-$Ca^{2+}$ ($\rho_{conc}$ =
0.70 and 0.47, respectively; the latter coefficient is higher for loads at 0.56; Table 5), suggesting that they
share some non-marine source(s). Furthermore, all glaciers have greater $Ca^{2+}$:$Mg^{2+}$ ratios than seawater
(0.32; Figure 5; Millero et al., 2008). It is likely that the excess $Ca^{2+}$ and $Mg^{2+}$ come from mineral particles,
i.e., $CaCO_3$ (calcite) and $CaMg(CO_3)_2$ (dolomite), derived from local rock (or soil) dust (Kekonen et al.,
2005), especially limestone, dolostone and marble, which are abundant in Svalbard (Dallmann, 1999). The
presence of carbonate ions in the collected snow samples would explain the missing negative charge in the
ionic balance (anion $X^-$; Figure S1).

Sulphate ($SO_4^{2-}$) is highly and significantly correlated ($p < 0.05$) with both $Na^+$ ($\rho_{load} = 0.92$; $\rho_{conc} = 0.80$)
and $Cl^-$ ($\rho_{load} = 0.93$; $\rho_{conc} = 0.75$), indicating that sea spray is its main source (Table 5). However, $Na^+/SO_4^{2-}$
and $Cl^-/SO_4^{2-}$ ratios are well below seawater values (Millero et al., 2008) on most glaciers except for those
near Hornsund (WB and HB), suggesting input of nss-$SO_4^{2-}$ (Figure 5). Biogenic nss-$SO_4^{2-}$ can occur in
snow as an oxidized by-product of dimethyl sulphide (DMS) emitted by marine algal blooms (Gondwe et

al., 2003), typically initiated in April but sometimes later (Ardyna et al., 2013). Another plausible source of nss-$SO_4^{2-}$ deposition in Svalbard is long-range atmospheric transport of secondary aerosols containing $SO_4^{2-}$, such as ammonium sulfate. This sulphate can be formed by $SO_x$ emitted from coal combustion throughout the winter and biomass burning in the spring (Barrie, 1986; Law and Stohl, 2007; Nawrot et al., 2016). The nss-$SO_4^{2-}$ does not correlate significantly with other ionic species, suggesting a separate origin. However, we need to caution that in the southern region of Svalbard, the estimation of ss-$SO_4^{2-}$ is subject to higher uncertainty because of the higher amount of $Na^+$ in the atmospheric deposition there.

Bulk ionic loads of $SO_4^{2-}$ in the snowpits are significantly and positively correlated with those of $NO_3^-$ ($\rho_{load}$ = 0.55) and $NH_4^+$ ($\rho_{load}$ = 0.68), but the correlations between weighted mean ionic concentrations are not significant, hinting at co-deposition (wet) rather than shared sources (Table 5). These species are known to form secondary aerosols (Karl et al., 2019; Schaap et al., 2004) and thus their proportions in aerosols may differ significantly from those in their source emissions. It is also possible that nitrogen species underwent further post-depositional photochemical reduction and evasion, thereby reducing their concentrations in snow (Curtis et al., 2018). Finally, we remark here that the snowpit sampling was done in April, earlier than the beginning of the oceanic algal bloom in the surrounding Svalbard basin, which could have led to an underrepresentation of biological emissions from late spring in our samples.

Spatial variations of ammonium load ($NH_4^+$) across Svalbard glaciers mirror the pattern shown by sea salt ions, with higher loads in the Hornsund area and lower loads in other areas. This is also reflected by significant correlations (Table 5) of the bulk loads of $NH_4^+$ with those of $Na^+$ and $Cl^-$ ($\rho_{load}$ = 0.64 and 0.73, respectively), and with $Na^+$, $K^+$ and $Mg^{2+}$ by concentration ($\rho_{conc}$ = 0.47, 0.62 and 0.47, respectively). Ammonium has been linked to biogenic, forest fire, and anthropogenic agricultural emissions (Trachsel et al., 2019). The higher annual snowpack load of $NH_4^+$, determined in the Hornsund area is more likely connected with biological sources than anthropogenic activities, although some contribution from biomass burning events cannot be excluded. The marine primary productivity in spring 2016 (April and May) was higher in the south-eastern ocean sector of the Svalbard archipelago (Figure S2), which could partially explain the higher $NH_4^+$ load. This would also explain the correlation between ammonium and sea-salt ions (Table 5). Locally, especially for HB, there may be extra $NH_4^+$ emissions from bird colonies (Keslinka et al., 2019; Wojczulanis K. et al., 2008).

Unlike $NH_4^+$, the bulk loading of $NO_3^-$ in snow is highest in northwestern Spitsbergen (Ny-Ålesund area), when compared to other parts of Svalbard. Deposition of $NO_3^-$ in Arctic snow is often associated with the long-range atmospheric transport of $NO_x$ and related N species from anthropogenic source regions at lower latitudes (Björkman et al., 2014; Fibiger et al., 2016; Vega et al., 2015a). Differences in $NO_3^-$ loads in snow in various parts of Svalbard might therefore reflect differences in the transport pathways of precipitating air masses, including the formation of secondary aerosols, or post-depositional processes, rather than local

emissions. While local shipping routes and the settlement of Ny-Ålesund itself may contribute $NO_3^-$
emissions (Winther et al., 2014), the highest share of the total ionic load of $NO_3^-$ was found in the
accumulation zone of HDF (9% of the total ionic load; Figure 3). Given that HDF is the most remote site
from Ny-Ålesund relative to KVG or ALB, it should not capture a high share of local pollution. The highest
correlation coefficient for $NO_3^-$, both in terms of concentrations and loads, was found with nss-$Ca^{2+}$. This
would support both the formation of calcium nitrate in the atmosphere (Gibson et al., 2006) or post-
depositional processes removing the $NO_3^-$ from layers poor in $Ca^{2+}$, since calcium has been hypothesised
to stabilise the nitrate in the snowpack against post-depositional losses (Kekonen et al., 2017).

*4.2. Chlorine depletion*
Although $Na^+$ and $Cl^-$, the main species of sea salt, are significantly correlated ($\rho_{conc}$ = 0.95, Table 5), the
values of the $Cl^-/Na^+$ ratio (1.8 w/w) in snow are lower than that in seawater on most studied glaciers,
except those near Hornsund (Figure 5), suggesting a $Cl^-$ deficit at the more northerly sites. Whillow et al.
(1992) found an opposite situation in the snowpack of Greenland, indicating $Cl^-/Na^+$ values higher than
the ratio of seawater. This $Cl^-$ enrichment relative to the $Cl^-/Na^+$ ratio in seawater may reflect Cl derived
from anthropogenic sources as well as from gas phase chlorine transportation and deposition in central
Greenland.
Contrastingly, a possible explanation of $Cl^-$ deficit in the Svalbard snowpack might be de-chlorination of
the sea spray aerosol during transport or, less likely, at the snow-atmosphere interface. This reaction occurs
between sea salt particles, containing NaCl, and $HNO_3$, $H_2SO_4$, or organic acids to release gaseous HCl
(Zhuang et al., 1999). We calculated the percentage of $Cl^-$ depletion ($Cl^-_{dep}$) as $Cl^-_{dep}$ = ($Cl^-_{ss}$ - $Cl^-_{meas}$) / $Cl^-$
$_{ss}$ × 100%, where $Cl^-_{ss}$ = 1.174 $Na^+_{meas}$, and $Cl^-_{meas}$ and $Na^+_{meas}$ are the measured equivalent concentrations
(Yao et al., 2003). Except for site HDF2 ($Cl^-_{dep}$ = 2%), the lowest mean $Cl^-_{dep}$ values were obtained for
Hornsund glaciers (WB, HB: 10–19%), while values at other glacier sites ranged between 21 and 75%
(Table 2). This suggests that sea-salt aerosols travel along a route from southern to northern Svalbard,
which gives more time for $Cl^-$ depletion in the ionic mixtures reaching more northerly locations.

*4.3. Bromine enrichment*
In addition to $Cl^-$, snowfall can scavenge $Br^-$ (Peterson et al., 2019; Spolaor et al., 2019). $Br^-$ loads on
Svalbard glaciers surveyed in April 2016 are positively and significantly correlated with those of primary
sea salt ions $Na^+$ ($\rho_{load}$ = 0.48), $Cl^-$ ($\rho_{load}$ = 0.53) and $Mg^{2+}$ ($\rho_{load}$ = 0.51) (Table 5). Correlations between
weighted mean concentrations are not significant, however, suggesting departures of the $Br^-$ concentrations
in snow from typical seawater ionic ratios at some glacier sites. A $Br^-$ enrichment factor ($Br_{enr}$) can be
calculated as $Br_{enr}$ = $Br^-$ / (0.0065 $Na^+$), where 0.0065 is the $Br^-$ : $Na^+$ seawater mass ratio (Maffezzoli et
al., 2017). The $Br_{enr}$ reflects specific processes (in particular sea ice Br emission) that affect the $Br^-$
concentration and load in the snowpack (Spolaor et al., 2014). Results of our calculations (Table 2, Figure
S3) show that for glaciers of the Hornsund area (HB and WB) and NW Spitsbergen (KVG, ALB and HDF),
the mean $Br_{enr}$ values are often $< 1$, indicating some $Br^-$ depletion, in agreement with the findings of Jacobi
et al (2019) for glaciers in the Ny-Ålesund area. The depletion could be a result of snowpack Br re-
emission, but this seems unlikely since field measurements near Ny-Ålesund found no evidence of such
volatilization of snow-bound Br (Spolaor et al., 2019).
Alternatively, $Br^-$ depletion could occur through BrO loss from marine aerosols and subsequent deposition
of these Br-depleted aerosols in snow. In contrast to southern and northwestern Spitsbergen, glaciers in
central Spitsbergen (LF) and in Nordaustlandet (AF), show $Br_{enr}$ values $> 1$. These glaciers lie relatively
close to areas to the east of the archipelago that are often covered by first-year sea ice. Newly-formed sea
ice has been shown to release gas phase Br into the polar atmosphere, thus supplying an extra Br source in
addition to sea spray (Spolaor et al., 2016). The spatial distribution of the Br-enriched snowpit sites
supports this, i.e., the sites closest to the areas covered by first-year sea ice have the largest Br enrichments,
and the latter decrease with greater distance from the eastern shores of Svalbard (Figure S3). A survey of
the average sea-ice coverage in the period March – May 2016, which is relevant to the Br enrichment
phenomenon, confirms that the north-eastern and eastern shore of Svalbard were indeed covered much
more frequently by close and open drift ice than the south or north-west (https://cryo.met.no/en/sea-ice).

### 4.4 Distribution pattern of $\delta^{18}O$ and $\delta^2H$

As described earlier, the SWE-weighted mean $\delta^{18}O$ and $\delta^2H$ values in glacier snowpits decrease
significantly with increasing latitude across Svalbard, the least negative values occurring on glaciers of the
Hornsund area, and the most negative in Austfonna (Table 3). This pattern follows the climate gradient
across the archipelago, milder in the south, colder in the north. Part of the south-north contrast in $\delta$ values
can be explained by the lower mean altitude of glacier sites in the Hornsund area compared to some of the
higher-elevation sites further north on Spitsbergen or on Austfonna. The relationship with elevation is
similar for both isotopic ratios in the collected dataset, with except AF that the isotopic signals might be
influenced by additional processes since it is an isolated ice cap mainly surrounding from ocean or sea ice
in winter.
Deuterium excess ($d = \delta^2H - (8 \cdot \delta^{18}O)$) is mainly influenced by the source region of the precipitating moisture
and in particular by the sea surface temperature, but also relative humidity and wind speed (Gat, 1996;
Uemura et al., 2008). In addition, $d$ is also influenced by the temperature gradient between the moisture
source and precipitation area (Johnsen et al., 1989). The SWE-weighted mean $d$ values in Svalbard
snowpits vary within a relatively narrow range of 6.74‰ (from 10.10 to 16.84 ‰), and similarly to $\delta^{18}O$,
show no clear gradient with elevation or longitude (Table 3). Deuterium excess shows a significant
correlation with latitude, at $\rho = 0.60$ ($p < 0.01$). A more detailed analysis of $d$ by latitude shows that
significantly different values are only obtained in snow pits sampled beyond 79.2 °N, i.e., in Austfonna
snowpits. This is confirmed by the Kruskal-Wallis test, i.e. rank-based ANOVA, calculated with two
groups of d values divided by the latitude threshold 79.2°N ($z = 4.23$, $p < 0.04$). In fact, drawing the latitude
threshold anywhere between 78.7 and 79.7 °N, a statistically significant difference with $p < 0.05$ is
obtained. This is consistent with lower temperatures and evaporation rates in the more northern waters
around Svalbard, and suggests that snowfall on AF is at least partly affected by a different, more northerly
moisture source than the rest of the archipelago.

*4.5 Effect of elevation: a case study of Na*

The glacier survey carried out during the C2S3 project afforded the opportunity to investigate the possible
effect of elevation on the ionic composition of the snowpack. To do this, we compared the bulk load and
SWE-weighted mean concentration of $Na^+$ across all studied snowpits, ordered by elevation (Figure 6).
Overall, both $Na^+$ load and concentration decrease with increasing altitude ($\rho_{load}$ = -0.24, not significant;
$\rho_{conc}$ = -0.72, p < 0.05). This likely reflects greater local sea spray aerosol deposition at lower, compared
to higher glacier sites. We then computed linear (Pearson) correlation coefficients ($R$, with associated *p*-
values) between log-transformed $Na^+$ loading ($\log(Na_{load})$) and $\delta^{18}O$ for all snowpits in the accumulation
zones of all seven glaciers (Figure 7). The calculation was performed with all snow layers. The $Na^+$ load
was used as sea-spray tracer, while the $\delta^{18}O$ was assumed to vary with moisture source between discrete
snowfall events. We find that the positive correlation between $\log(Na_{load})$ and $\delta^{18}O$ increases with elevation
from $R$ = 0.1 (HB3; 396 m a.s.l.) to $R$ = 0.65 (LF3; 1193 m a.s.l.), and reaches a 95 % threshold of
significance ($R$ > 0.3) for glacier sites above 600 m a.sl. (KVG, AF, LF and HDF; Figure 7). The average
distance from the sea is a comparatively negligible factor in explaining the correlation between $\log(Na_{load})$
and $\delta^{18}O$.

The increase in strength and significance of the $\log(Na_{load})$-$\delta^{18}O$ correlation with altitude might be
explained by different contributions of locally emitted $ssNa^+$, relative to those of $Na^+$ from more distant
sources. Sites located at lower altitudes are proportionally more affected by local sea spray deposition,
with or without snowfall. Conversely, sites at higher elevations likely receive a larger share of their ionic
load from more distant sources, and by wet deposition through snowfall. At the four sites (KVG, AF, LF,
and HDF) where the $\log(Na_{load})$-$\delta^{18}O$ correlation is significant, increases in $\delta^{18}O$ in snow layers are often
associated with higher $Na^+$ concentrations. It is rather difficult to propose a precise explanation for this
association. However, we would indicate that the isotopically heavier (less negative) $\delta^{18}O$ values suggest
that the co-registered $Na^+$ enhancements are associated with precipitation of relatively warm air event,
probably advected from lower latitudes. The snowfall associated with a warm event is able to wet scavenge
the sea spray aerosol present in the atmosphere. On the contrary, when cold air masses (Arctic type)
dominate, snowfall events are relatively limited due to the low air humidity causing a lower efficiency of
wet scavenging. This results in lower $\delta^{18}O$ and (likely) Na sodium loads, suggesting that wet deposition
dominates the chemical load of the snowpack. Although this process should occur also at lower elevation
sites, the local emission and associated dry deposition are likely more important than wet deposition. More
frequent melt-refreeze episodes at lower elevations would also mask the proposed relationship (as
suggested by the vertical profiles of stratigraphy reported in Figure S4).
Another possible explanation is that in the Arctic, air masses are transported from low to high elevation
sites without any strong disturbance of the atmospheric conditions. In this case, isotopically heavier
molecules and sea spray particles are gradually scavenged from the air masses. If this was the main process,
we should find the correlation across all studied sites, assuming that $Na^+$ scavenged at a similar rate as that
of isotopic fractionation. Since this has not been found, we propose that the correlation at higher elevation
cannot be explained by atmospheric distillation alone. The possibility that the correlation is due to different
sources of air masses seems unsupported due to the absence of a correlation between d-excess and sodium.

**5. Summary and Conclusion**
We have quantified and described, for the first time, the spatial distribution of major ion loads ($Ca^{2+}$, $K^+$,
$Na^+$, $Mg^{2+}$, $NH_4^+$, $SO_4^{2-}$, $Br^-$, $Cl^-$ and $NO_3^-$) and variations of $\delta^{18}O$ and $\delta^2H$ in the snowpack on glaciers
across Svalbard for a single accumulation season (2015-2016). The highest total ionic loads are found in
the southern region of Spitsbergen (Hornsund area), and exceed 8 g m$^{-2}$. Conversely, the lowest total ionic
loads ($\leq$ 0.6 g m$^{-2}$) are found at sites in central or northwestern Spitsbergen (LF and HDF). Sea salt ions
($Cl^-$, $Na^+$, and $SO_4^{2-}$) dominate the ionic loads at all sites, but their share is highest at sites near Hornsund,
for, e.g., 48% $Cl^-$, compared to only 29% on Holtedahlfonna. Relatively elevated $Ca^{2+}/Mg^{2+}$ ratios in snow
at all sites indicate non-sea-salt $Ca^{2+}$ inputs, most likely in the form of carbonate dust. Unlike other ions,
$NO_3^-$ has the highest loads in glaciers of northwestern Spitsbergen, and the lowest at LF. The nitrogen
species, $NO_3^-$ and $NH_4^+$, show distinct spatial distribution patterns. The highest $NO_3^-$ loads are found in the
northwestern part of Svalbard, while the highest $NH_4^+$ loads are in the southwest. Bromide ($Br^-$) is most
enriched in snow relative to seawater at AF and LF, the glacier sites located closest to areas with first-year
sea ice cover. This supports first-year sea ice being an important source of non-sea salt $Br^-$ in the polar
atmosphere.

An increasing positive correlation between $\log(Na_{load})$ and $\delta^{18}O$ as a function of elevation sites suggests
that locations above 600-700 m a.s.l. are influenced by a proportionally higher share of ions from distant
sources, while the lower sites are exposed to more local sources, especially sea spray. These findings
confirm that the optimal sites to study the effects of long-range pollution deposition in Svalbard are those
at higher elevations, such as the accumulation zones of HDF or LF, because they are the sites least impacted
by local aerosol emissions. The current study gives the first picture of the ionic composition in the Svalbard
snowpack in different regions across the archipelago, in the context of which processes are relevant in
controlling the annual snowpack chemical composition, especially the influence of local and long-range
transport.


**Acknowledgements**

The work developed here was supported through grants 246731/E10 and 257636/E10 from the Svalbard Science Forum /Research Council of Norway, by the Gothenburg Centre of Advanced Studies, BECC - Biodiversity and Ecosystem Services in a Changing Climate, Gothenburg Air and Climate Centre, International Arctic Science Committee - Cryosphere working group and the Norwegian Polar Institute. Part of fieldwork has been conducted thanks to the funds of the Leading National Research Centre (KNOW) in Poland, received by the Centre for Polar Studies for the period 2014–2018. This research was also partially supported within statutory activities No 3841/E-41/S/2020 of the Ministry of Science and Higher Education of Poland. The project has received further funding from the European Union's Horizon 2020 research and innovation programme under grant agreement no. 689443 via project iCUPE (Integrative and Comprehensive Understanding on Polar Environments).

**Author Contribution**

EB, KK and AS wrote the manuscript, with contributions from all co-authors. JCG, MB, CL, BL, TS, CZ, FL, DK, AS, EB, TM, KK and AU initiated the April 2016 survey. EB, KK CPV and CZ perform the analytical measurements, TM the $\delta18O$ analyses.

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

   **FIGURES**

**Figure 1.** Total snowpack loads (mg m$^{-2}$) of major ions in 22 snowpits collected on glaciers during the C2S3
project. Seven glaciers were sampled in three snowpits in the lower ablation zone (1), near the equilibrium line
(2) and in the upper accumulation zone (3), except on KVG glacier where an extra snowpit was sampled within
the ablation zone.
Abbreviations: KVG = Kongsvegen, HDF = Holtedahlfonna, AF= Austfonna, ALB = Austre Lovénbreen, LF =
Lomonosovfonna, HB = Hansbreen, WB = Werenskiöldbreen.

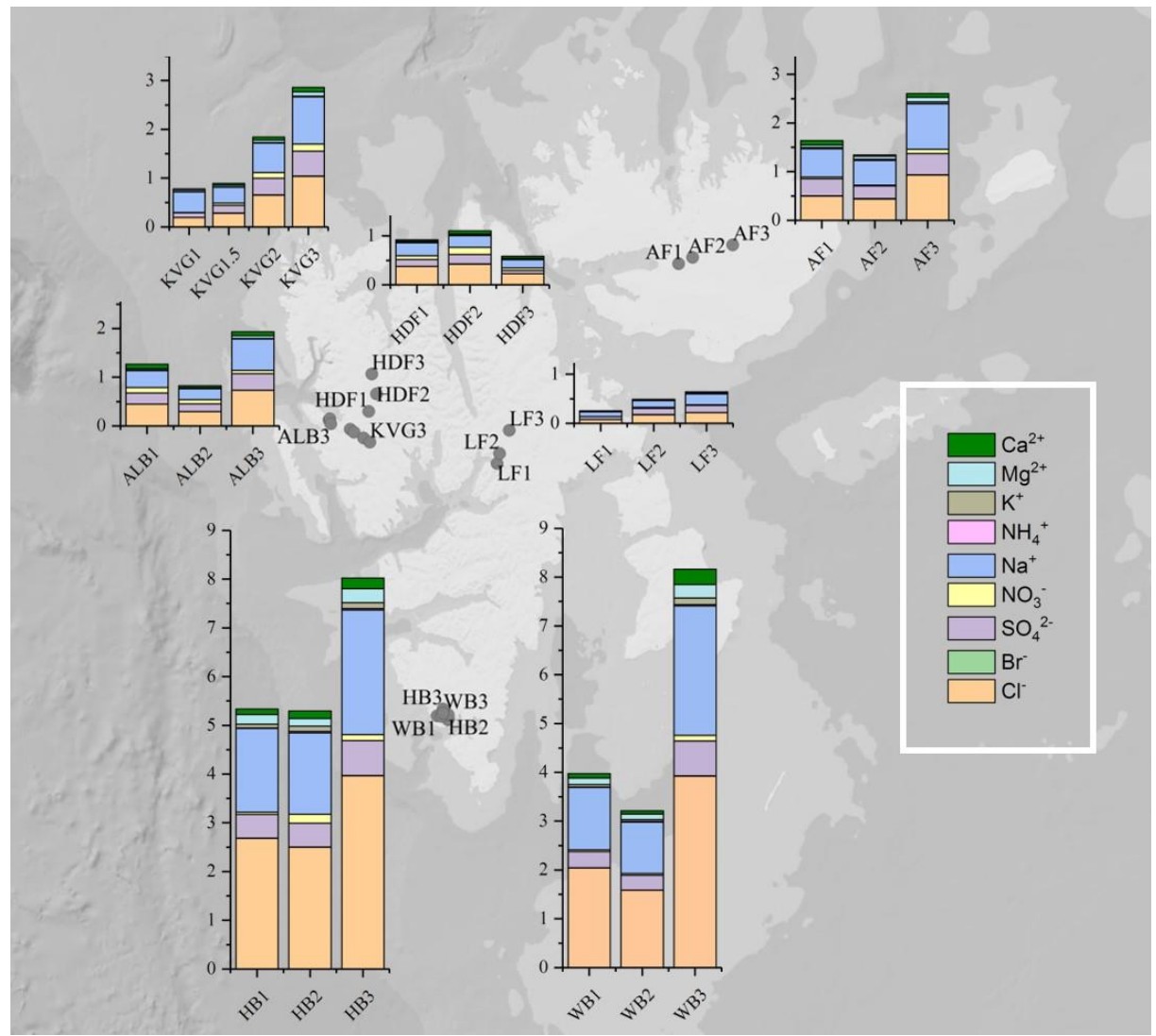


**Figure 2.** Calculated ionic loads in the snowpack (mg m$^{-2}$) at the 7 glacier sites sampled during the C2S3 project. Snowpits for each glacier are marked with the same colour and ordered from lower (left) to higher altitudes (right). For the KVG another snowpit was dug between glacier zones 1 and 2.

Abbreviation: KVG = Kongsvegen, HDF = Holtedahlfonna, AF= Austfonna, ALB = Austre Lovénbreen, LF = Lomonosovfonna, HB = Hansbreen, WB = Werenskiöldbreen.

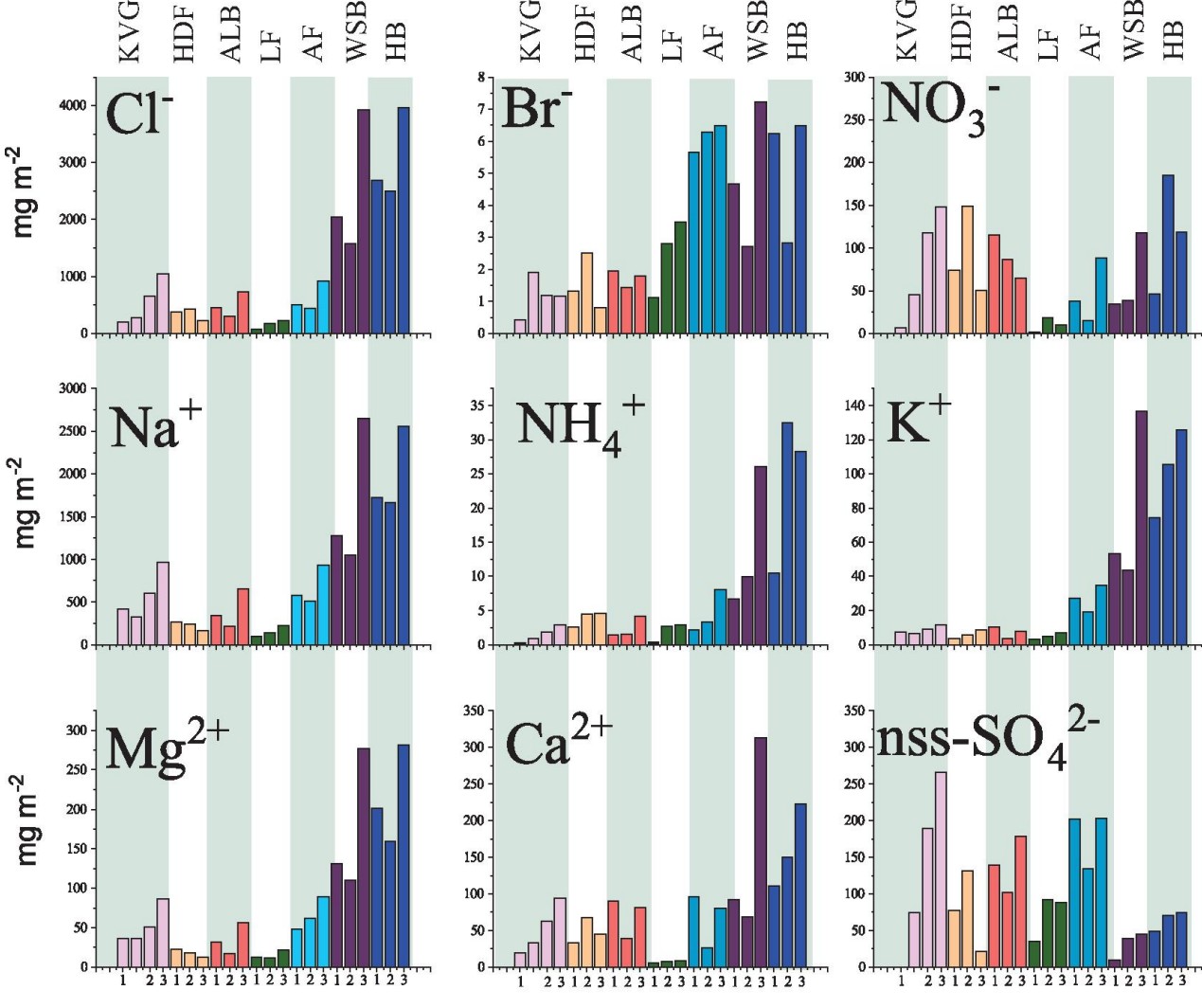

805 **Figure 3.** Pie diagrams showing relative ionic composition in the snowpits dug in the accumulation
806 zones of the studied glaciers.

807

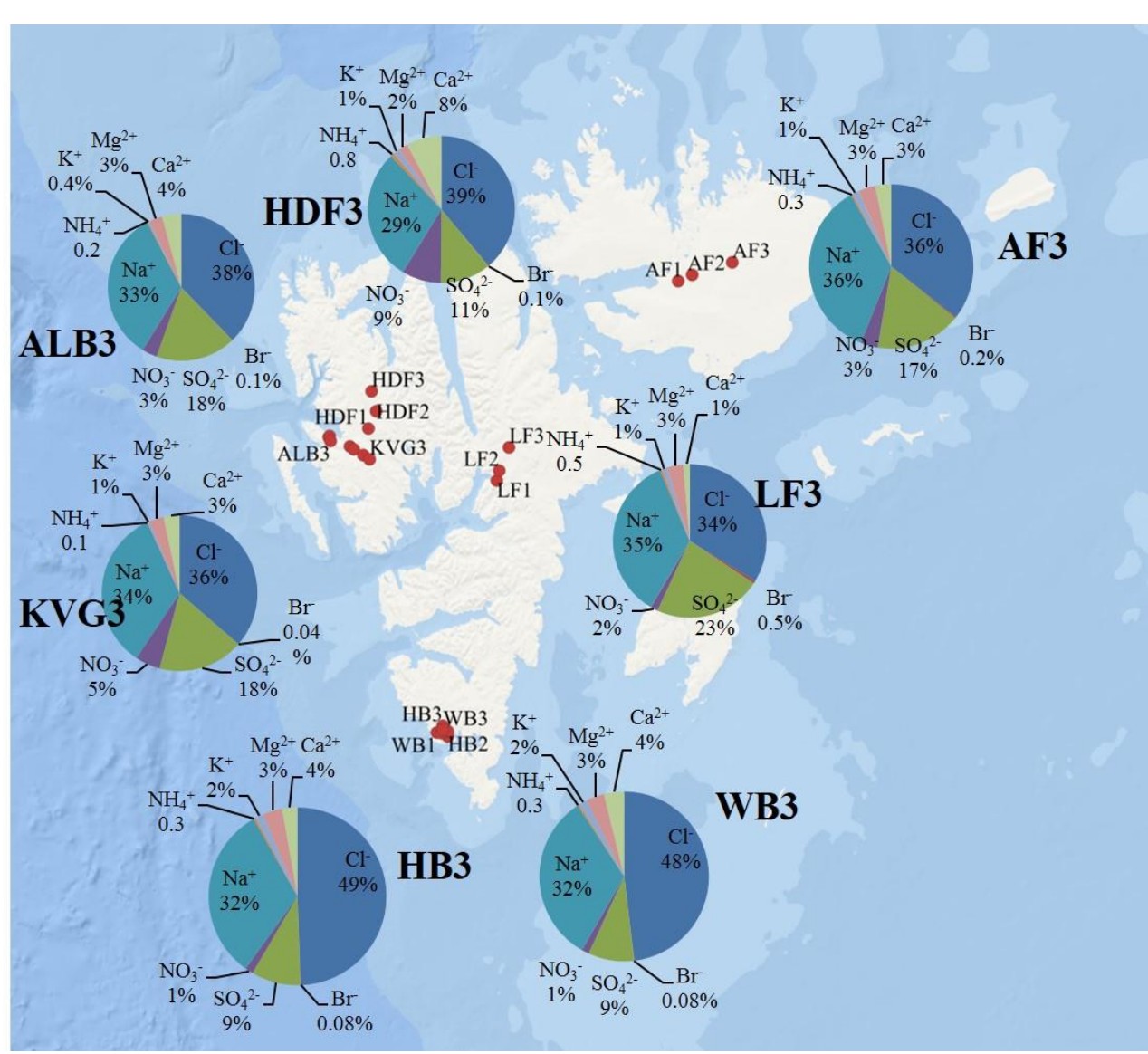

**Figure 4**. Box plots of stable water isotopes ($\delta^{18}O$ and $\delta^{2}H$) and deuterium excess ($d$) for each
snowpit.

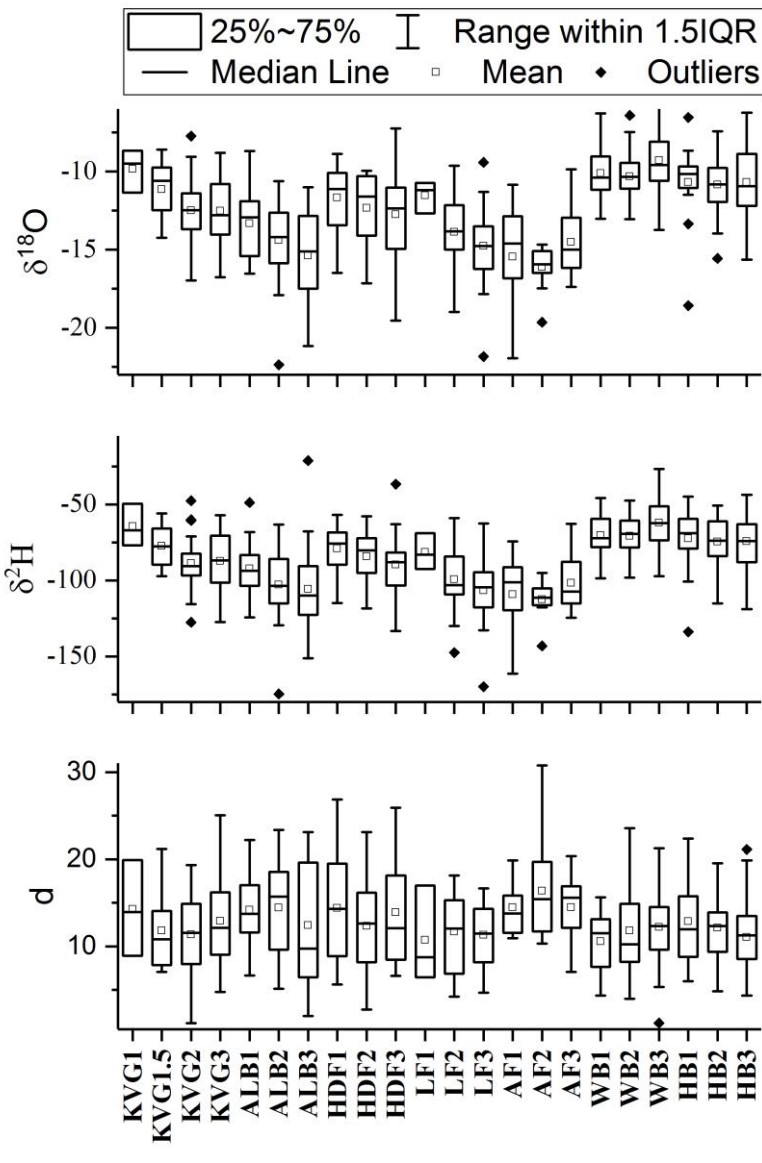








**Figure 5.** Panels from top: 1) $Cl^-/Na^+$; 2) $Na^+/SO_4^{2-}$; 3) $Cl^-/SO_4^{2-}$; 4) the total loads of sea-salt sulphate (ss-
$SO_4^{2-}$) and non-sea-salt sulphate (nss-$SO_4^{2-}$), and 5) $Ca^{2+}/Mg^{2+}$ for all glaciers investigated during the C2S3
project (in spring 2016).


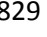


**Figure 6.** Sodium load in snowpits ordered by increasing elevation in m a.s.l., indicated by the red lines. The
colours identify areas where the snowpits have been excavated: each colour represents a separate glacier (HB
– blue; WB – purple; ALB – red; LF – green; KVG – pink; AF – light blue; HDF – orange). IQR = inter-
quartile range, i.e. the difference between the value of quartiles 3 and 1.

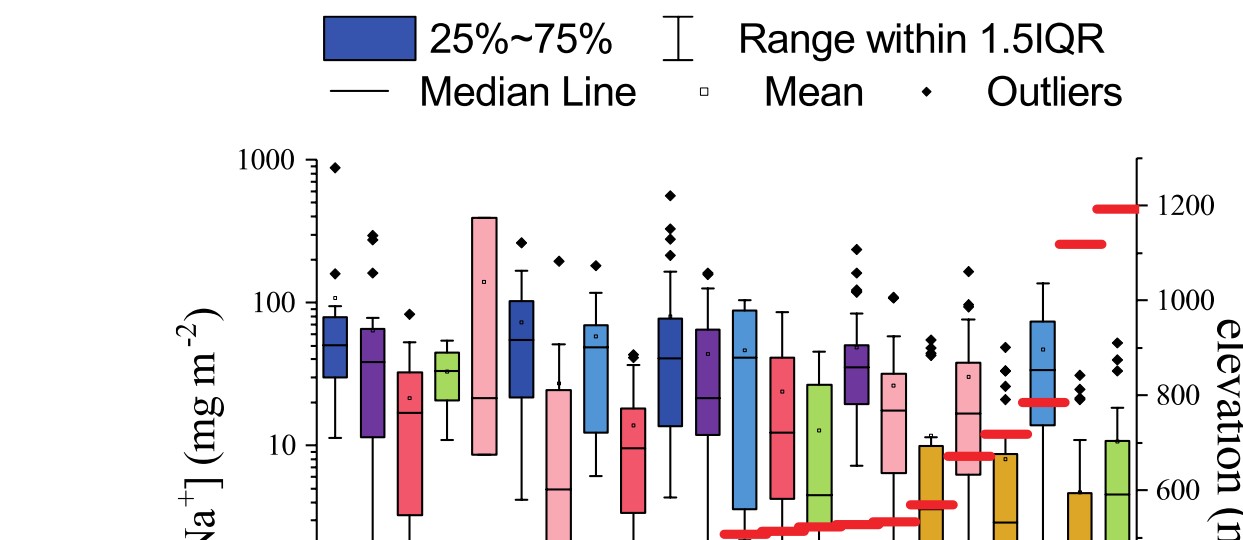


**Figure 7.** The correlation coefficient between the oxygen isotope ratio ($\delta^{18}O$) and log[Na$_{load}$] increases with
elevation. The left axis represents the correlation coefficient (R) between log[Na$_{load}$] and $\delta^{18}O$, using the entire
dataset for each snowpit (i.e. all layers have been used for the statistical correlation). The *x axis* indicates the
altitude of the snowpit. The upper panel shows the p-value: correlations have been considered statistically
significant if $p < 0.05$.

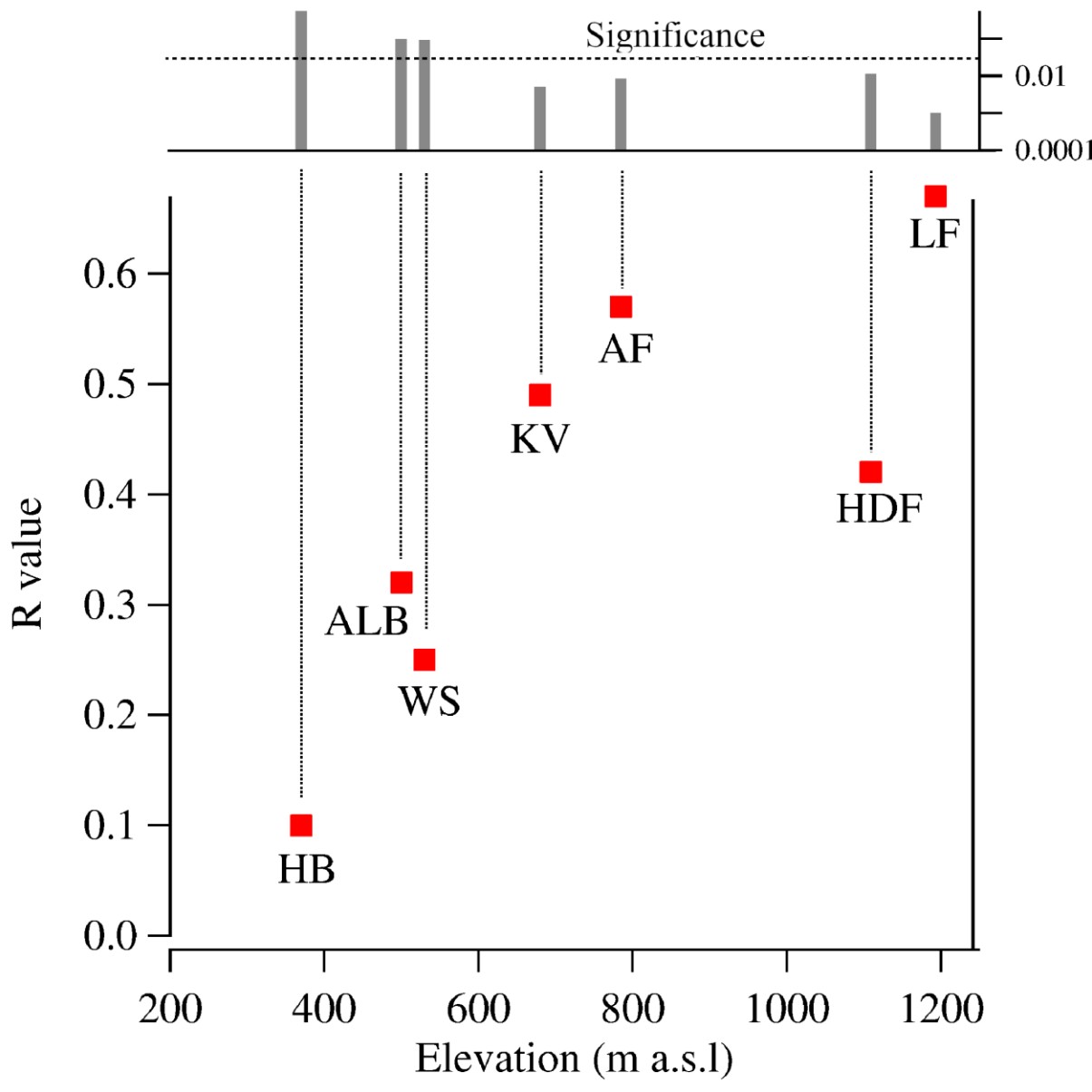


**TABLES**
**Table 1.** Table 1. Glaciers and sampling sites included in this study with their main characteristic. The air
temperature was measured with a digital thermometer when the operators started to dig the snowpits. AWS:
atmospheric weather station; UiO: University of Oslo; ThèMA: Thèoriser & Modèliser pour Amènager,
University of Franche-Comté; NPI: Norwegian Polar Institute; IMAU: Institute for Marine and Atmospheric
Research, Utrecht University; UoS: University of Silesia, IG PAS – Institute of Geophysics, Polish Academy
of Sciences; CNR – Consiglio Nazionale delle Ricerche. Seven glaciers were considered at three different
altitudes: 1) lower ablation zone; 2) ELA; 3) upper accumulation zone. Exceptionally, two snowpits (KVG 1
and KVG 1.5) were dug in the ablation zone of Kongsvegen glacier.

**Table 2.** Total load (mg m$^{-2}$) of major ions, calculated as the sum of loads in all layers of each snowpit. Sea
salt sulphate (ss-$SO_4^{2-}$) and non-sea-salt sulphate (nss-$SO_4^{2-}$) are expressed as mg m$^{-2}$, while chloride depletion
($Cl_{dep}^-$) is given as a percentage and bromide enrichment ($Br_{enr}$) refers to an enrichment compared to seawater
composition.

**Table 3.** Volume-weighted mean concentrations of major ions in each snowpit (calculated as the sum of loads
in all layers divided by the total SWE of the snowpit): the nss (non-sea-salt) fractions have been calculated in
each layer before the volume-weighting procedure. Average SWE-weighted stable water isotope ratios ($\delta^{18}O$
and $\delta^2H$ expressed as ‰) and average deuterium excess (*d*) are also reported.

**Table 4.** Loads (mg m$^{-2}$) of selected major ions from the 2016 sampling and from earlier studies at
Lomonosovfonna summit (LF3), corresponding to the concentrations given in Fig. 4.

**Table 5.** Spearman rank order correlations of a) ionic loads (mg m$^{-2}$) and b) SWE-weighted mean
concentrations of major ions across all 7 glaciers (n = 22 locations). ns = non-significant correlations (*p*-value
> 0.05). Ionic loads were calculated from all snowpit layers, while SWE-weighted mean concentrations were
calculated by dividing the total ionic loads in each snowpit by its total SWE. Non-sea-salt (nss) components
were estimated based from seawater ratios: $Ca^{2+}/Na^+$ is 0.038 while $SO_4^{2-}/Na^+$ is 0.252 (w/w; Millero et al.,

888 2008).


**Table 1.** Glaciers and sampling sites included in this study with their main characteristics. The air temperature was measured with a digital thermometer when the operators started to dig the snow pits. AWS: automatic weather station; UiO: University of Oslo; ThèMA: Thèoriser & Modèliser pour Amènager, Université de Franche-Comté; NPI: Norwegian Polar Institute; IMAU: Institute for Marine and Atmospheric Research, Utrecht University; UoS:  University of Silesia, IG PAS – Institute of Geophysics, Polish Academy of Sciences; CNR – Consiglio Nazionale delle Ricerche.

| Glacier | Site | Zone | AWS | Lat. (°N) | Lon. (°E) | Elev. (m) | Date (dd.mm.yyyy) | Air Temp. (°C) | Snow height (cm) | Snow Water Equivalent (SWE) (mm) |
|---|---|---|---|---|---|---|---|---|---|---|
| | AF1 | ablation | | 79.734 | 22.414 | 336 | 21.04.2016 | -13.5 | 106 | 330.50 |
| Austfonna | AF2 | equilibrium line | UiO | 79.767 | 22.825 | 507 | 23.04.2016 | -7.1 | 135 | 439.59 |
| | AF3 | accumulation | | 79.832 | 24.004 | 785 | 24.04.2016 | -14.7 | 181 | 803.93 |
| | ALB1 | ablation | ThéMA | 78.883 | 12.136 | 195 | 25.04.2016 | -3.7 | 81 | 296.66 |
| Austre Lovénbreen | ALB2 | equilibrium line | \CNR | 78.889 | 12.159 | 340 | 25.04.2016 | -2.8 | 90 | 353.17 |
| | ALB3 | accumulation | \NPI | 78.861 | 12.187 | 513 | 20.04.2016 | -11.3 | 161 | 499.67 |
| | KVG1 | ablation | | 78.830 | 12.759 | 226 | 13.04.2016 | -13.9 | 20 | 51.29 |
| | KVG1.5 | ablation | | 78.813 | 12.869 | 326 | 13.04.2016 | -13.9 | 75 | 261.94 |
| Kongsvegen | KVG 2 | equilibrium line | NPI\CNR | 78.780 | 13.153 | 534 | 11.04.2016 | -17.5 | 162 | 575.78 |
| | KVG3 | accumulation | | 78.756 | 13.336 | 672 | 12.04.2013 | -15.5 | 234 | 880.13 |
| | HDF1 | ablation | | 78.931 | 13.303 | 570 | 17.04.2016 | -14.5 | 108 | 372.98 |
| Holtedahlfonna | HDF2 | equilibrium line | NPI\CNR | 79.029 | 13.531 | 718 | 17.04.2016 | -14.2 | 175 | 625.00 |
| | HDF3 | accumulation | | 79.140 | 13.394 | 1119 | 15.04.2016 | -18.1 | 201 | 732.08 |
| | LF1 | ablation | | 78.633 | 17.077 | 223 | 10.04.2016 | -10.9 | 27 | 99.4 |
| Lomonosovfonna | LF2 | equilibrium line | IMAU | 78.691 | 17.150 | 523 | 9.04.2016 | -5.8 | 94 | 277.28 |
| | LF3 | accumulation | | 78.824 | 17.435 | 1193 | 11.04.2016 | -24 | 146 | 487.01 |
| | HB1 | ablation | | 77.049 | 15.639 | 102 | 25.04.2016 | -7.3 | 102 | 396.10 |
| Hansbreen | HB2 | equilibrium line | UoS/IG PAS | 77.083 | 15.639 | 275 | 25.04.2016 | -6.9 | 169 | 640.28 |
| | HB3 | accumulation | | 77.120 | 15.487 | 396 | 29.04.2016 | 0.7 | 288 | 1305.09 |
| | WB1 | ablation | | 77.075 | 15.313 | 166 | 16.04.2016 | -9.2 | 81 | 328.34 |
| Werenskiöldbreen | WB2 | equilibrium line | UoS | 77.072 | 15.441 | 413 | 16.04.2016 | -11.2 | 110 | 454.75 |
| | WB3 | accumulation | | 77.092 | 15.489 | 528 | 18.04.2016 | -11.1 | 330 | 1396.60 |

**Table 2.** Total load (mg m$^{-2}$) of major ions, calculated as the sum of loads in all layers of each snow pit. Sea-salt sulphate (ss-SO$_4^{2-}$) and non-sea-salt sulphate (nss-SO$_4^{2-}$) are expressed as mg m$^{-2}$, while chloride depletion (Cl$^-_{dep}$) is given as a percentage and bromide enrichment (Br$_{enr}$) refers to an enrichment compared to seawater.

| Site | Cl$^-$ | Br$^-$ | SO$_4^{2-}$ | NO$_3^-$ | Na$^+$ | NH$_4^+$ | K$^+$ | Mg$^{2+}$ | Ca$^{2+}$ | ss-SO$_4^{2-}$ | nss-SO$_4^{2-}$ | Cl$^-_{dep}$ % | Br$_{enr}$ |
|------|------|------|------|------|------|------|------|------|------|------|------|------|------|
| KVG1 | 190 | 0.4 | 96 | 6 | 421 | 0.3 | 7 | 37 | 20 | 106 | nd | 75 | 0.2 |
| KVG1.5 | 281 | 1.9 | 157 | 45 | 327 | 0.9 | 6 | 36 | 33 | 82 | 75 | 52 | 1.0 |
| KVG2 | 652 | 1.2 | 342 | 118 | 605 | 1.9 | 9 | 51 | 63 | 152 | 190 | 40 | 0.3 |
| KVG3 | 1039 | 1.2 | 509 | 148 | 967 | 2.9 | 12 | 86 | 94 | 244 | 266 | 40 | 0.2 |
| HDF1 | 373 | 1.3 | 144 | 74 | 267 | 2.6 | 4 | 23 | 33 | 68 | 79 | 21 | 0.9 |
| HDF2 | 423 | 2.5 | 192 | 148 | 240 | 4.5 | 6 | 18 | 67 | 61 | 131 | 2 | 1.7 |
| HDF3 | 227 | 0.8 | 65 | 51 | 170 | 4.5 | 9 | 12 | 45 | 43 | 22 | 25 | 0.8 |
| ALB1 | 446 | 1.9 | 226 | 115 | 343 | 1.5 | 10 | 31 | 90 | 86 | 139 | 27 | 0.9 |
| ALB2 | 294 | 1.4 | 158 | 87 | 221 | 1.6 | 4 | 17 | 38 | 56 | 107 | 25 | 1.2 |
| ALB3 | 729 | 1.8 | 342 | 64 | 648 | 4.2 | 8 | 56 | 81 | 158 | 165 | 36 | 0.4 |
| LF1 | 75 | 1.1 | 59 | 1 | 95 | 0.4 | 3 | 12 | 6 | 31 | 48 | 51 | 2.1 |
| LF2 | 174 | 2.8 | 127 | 18 | 141 | 2.6 | 5 | 12 | 7 | 53 | 127 | 38 | 3.2 |
| LF3 | 216 | 3.5 | 144 | 10 | 225 | 2.9 | 7 | 22 | 9 | 56 | 93 | 45 | 2.7 |
| AF1 | 498 | 5.7 | 348 | 38 | 578 | 2.2 | 27 | 48 | 95 | 127 | 173 | 53 | 1.6 |
| AF2 | 438 | 6.3 | 263 | 15 | 509 | 3.4 | 19 | 62 | 26 | 147 | 153 | 51 | 2.0 |
| AF3 | 928 | 6.5 | 439 | 88 | 933 | 8.1 | 35 | 89 | 81 | 185 | 206 | 54 | 1.5 |
| WB1 | 2041 | 4.7 | 332 | 34 | 1278 | 6.7 | 53 | 131 | 91 | 340 | 15 | 10 | 0.6 |
| WB2 | 1584 | 2.7 | 304 | 38 | 1051 | 9.9 | 44 | 110 | 68 | 220 | 24 | 16 | 0.4 |
| WB3 | 3922 | 7.2 | 713 | 118 | 2649 | 26.1 | 137 | 277 | 313 | 671 | 37 | 17 | 0.5 |
| HB1 | 2680 | 6.2 | 482 | 46 | 1722 | 10.5 | 74 | 201 | 110 | 475 | 47 | 13 | 0.6 |
| HB2 | 2499 | 2.8 | 490 | 185 | 1667 | 32.5 | 105 | 159 | 150 | 350 | 73 | 19 | 0.3 |
| HB3 | 3964 | 6.5 | 719 | 118 | 2557 | 28.2 | 125 | 281 | 223 | 751 | 107 | 13 | 0.4 |

**Table 3.** Volume-weighted mean concentrations of major ions in each snow pit (calculated as the sum of concentrations in all layers divided by the total SWE of the snow pit): the nss ( non-sea-salt) fractions were calculated in each layer before the volume-weighting procedure. Average SWE-weighted stable water isotope ratios ($\delta^{18}O$ and $\delta 2H$ expressed as ‰) and average deuterium excess (*d*) are also reported.

| Site | Cl⁻ | Br⁻ | SO₄²⁻ | NO₃⁻ | Na⁺ | NH₄⁺ | K⁺ | Mg²⁺ | Ca²⁺ | nss-SO₄²⁻ | nss- K⁺ | nss- Mg²⁺ | nss-Ca²⁺ | $\delta^{18}O$ | $\delta 2H$ | d |
|------|-----|-----|--------|------|-----|------|-----|------|------|-----------|---------|-----------|----------|---------|---------|-----|
| KVG1 | 3.71 | 0.008 | 1.88 | 0.12 | 8.21 | 0.005 | 0.15 | 0.716 | 0.384 | (-0.19) | (-0.16) | (-0.26) | 0.07 | -9.69 | -66.17 | 11.37 |
| KVG1.5 | 1.07 | 0.007 | 0.60 | 0.17 | 1.25 | 0.004 | 0.01 | 0.139 | 0.126 | 0.29 | (-0.02) | (-0.01) | 0.08 | -11.32 | -78.25 | 12.34 |
| KVG2 | 1.13 | 0.002 | 0.60 | 0.21 | 1.05 | 0.003 | 0.02 | 0.088 | 0.109 | 0.33 | (-0.02) | (-0.04) | 0.07 | -12.51 | -88.62 | 11.48 |
| KVG3 | 1.18 | 0.001 | 0.58 | 0.17 | 1.10 | 0.003 | 0.01 | 0.098 | 0.107 | 0.30 | (-0.03) | (-0.03) | 0.07 | -12.72 | -89.50 | 12.25 |
| HDF1 | 1.03 | 0.004 | 0.39 | 0.21 | 0.72 | 0.007 | 0.01 | 0.062 | 0.098 | 0.21 | (-0.02) | (-0.02) | 0.07 | -13.51 | -94.37 | 13.75 |
| HDF2 | 0.68 | 0.004 | 0.31 | 0.24 | 0.39 | 0.007 | 0.01 | 0.029 | 0.108 | 0.21 | (-0.01) | (-0.02) | 0.09 | -13.91 | -99.15 | 12.10 |
| HDF3 | 0.31 | 0.001 | 0.09 | 0.07 | 0.23 | 0.006 | 0.01 | 0.016 | 0.062 | 0.03 | 0.00 | (-0.01) | 0.05 | -15.18 | -104.51 | 16.97 |
| ALB1 | 1.50 | 0.007 | 0.76 | 0.39 | 1.16 | 0.005 | 0.04 | 0.106 | 0.304 | 0.47 | (-0.01) | (-0.03) | 0.25 | -11.22 | -75.17 | 14.59 |
| ALB2 | 0.84 | 0.005 | 0.46 | 0.27 | 0.63 | 0.005 | 0.01 | 0.049 | 0.116 | 0.30 | (-0.01) | (-0.03) | 0.09 | -12.19 | -83.11 | 14.40 |
| ALB3 | 1.43 | 0.003 | 0.65 | 0.12 | 1.25 | 0.006 | 0.01 | 0.107 | 0.161 | 0.33 | (-0.03) | (-0.04) | 0.11 | -12.40 | -85.40 | 13.79 |
| LF1 | 1.09 | 0.016 | 0.80 | 0.06 | 1.250 | 0.012 | 0.040 | 0.143 | 0.076 | 0.48 | (-0.006) | (-0.006) | 0.028 | -11.61 | -82.79 | 10.10 |
| LF2 | 0.84 | 0.015 | 0.65 | 0.07 | 0.753 | 0.013 | 0.027 | 0.065 | 0.044 | 0.46 | (-0.001) | (-0.024) | 0.015 | -14.54 | -105.44 | 10.90 |
| LF3 | 0.45 | 0.007 | 0.31 | 0.02 | 0.456 | 0.006 | 0.014 | 0.043 | 0.015 | 0.19 | (-0.003) | (-0.012) | -0.003 | -15.14 | -110.42 | 10.69 |
| AF1 | 1.28 | 0.014 | 0.91 | 0.10 | 1.524 | 0.005 | 0.070 | 0.110 | 0.278 | 0.52 | 0.013 | (-0.072) | 0.220 | -14.34 | -100.76 | 13.94 |
| AF2 | 1.16 | 0.016 | 0.68 | 0.03 | 1.331 | 0.008 | 0.052 | 0.170 | 0.069 | 0.35 | 0.003 | 0.012 | 0.018 | -16.00 | -111.15 | 16.84 |
| AF3 | 0.76 | 0.008 | 0.49 | 0.11 | 0.914 | 0.011 | 0.034 | 0.081 | 0.090 | 0.26 | 0.000 | (-0.028) | 0.055 | -13.89 | -96.89 | 14.24 |
| WB1 | 6.596 | 0.014 | 1.079 | 0.105 | 4.12 | 0.02 | 0.18 | 0.43 | 0.27 | 0.05 | 0.02 | (-0.05) | 0.11 | -10.17 | -70.62 | 10.75 |
| WB2 | 2.886 | 0.005 | 0.536 | 0.066 | 1.92 | 0.01 | 0.07 | 0.19 | 0.14 | 0.05 | 0.00 | (-0.04) | 0.07 | -10.25 | -70.14 | 11.90 |
| WB3 | 2.824 | 0.005 | 0.506 | 0.086 | 1.91 | 0.02 | 0.10 | 0.20 | 0.17 | 0.03 | 0.03 | (-0.03) | 0.10 | -9.54 | -63.64 | 12.66 |
| HB1 | 7.378 | 0.016 | 1.316 | 0.127 | 4.76 | 0.03 | 0.20 | 0.57 | 0.49 | 0.12 | 0.02 | 0.01 | 0.31 | -11.14 | -75.93 | 13.19 |
| HB2 | 3.155 | 0.004 | 0.661 | 0.283 | 2.17 | 0.04 | 0.12 | 0.19 | 0.21 | 0.11 | 0.04 | (-0.07) | 0.13 | -10.69 | -73.34 | 12.17 |
| HB3 | 3.573 | 0.005 | 0.658 | 0.098 | 2.28 | 0.03 | 0.12 | 0.26 | 0.19 | 0.08 | 0.04 | (-0.02) | 0.10 | -11.25 | -77.62 | 12.35 |

**Table 4.** Loads (mg m$^{-2}$) of selected major ions from the 2016 sampling and from earlier studies at Lomonosovfonna summit (LF3), corresponding to the concentrations given in Fig. 4.

| Year | Na$^+$ | Ca$^{2+}$ | NO$_3^-$ | nss-SO$_4^{2+}$ | Study |
|------|--------|-----------|----------|-----------------|-------|
| 2002 | 126.7 | 7.1 | 27.3 | 37.1 | (Virkkunen et al., 2007) |
| 2009 | n.a. | n.a. | 33.5 | n.a. | *Vega C. (unpublished data)* |
| 2010 | 80.1 | 24.3 | 52.3 | 48.1 | *Vega C. (unpublished data)* |
| 2011 | 262.9 | 46.2 | 27.2 | 34.1 | (Vega et al., 2015), *Vega C.(unpublished data)* |
| 2016 | 222.2 | 7.2 | 11.4 | 93.0 | *This study* |

**Table 5.** Spearman rank order correlations of a) ionic loads (mg m$^{-2}$) and b) SWE-weighted mean concentrations of major ions across all 7 glaciers (n=22 locations). ns = non-significant correlations ($p$-value $> 0.05$). Ionic loads were calculated from all snow pit layers, while SWE-weighted mean concentrations were calculated by dividing the ionic loads in each snow pit by its total SWE. Non-sea-salt (nss) components were estimated based from seawater ratios: Ca$^{2+}$/Na$^{+}$ is 0.038 while and SO$_4^{2-}$/Na$^{+}$ is 0.252 (w/w; Millero et al., 2008).

a)

| | Cl$^-$ | Br$^-$ | SO$_4^{2-}$ | NO$_3^-$ | Na$^+$ | NH$_4^+$ | K$^+$ | Mg$^{2+}$ | Ca$^{2+}$ | nss-SO$_4^{2-}$ |
|---|---|---|---|---|---|---|---|---|---|---|
| Br$^-$ | 0.53 | | | | | | | | | |
| SO$_4^{2-}$ | 0.93 | 0.60 | | | | | | | | |
| NO$_3^-$ | 0.55 | ns | 0.55 | | | | | | | |
| Na$^+$ | 0.94 | 0.48 | 0.92 | 0.44 | | | | | | |
| NH$_4^+$ | 0.73 | 0.62 | 0.68 | ns | 0.64 | | | | | |
| K$^+$ | 0.82 | 0.61 | 0.81 | ns | 0.85 | 0.75 | | | | |
| Mg$^{2+}$ | 0.90 | 0.51 | 0.88 | ns | 0.98 | 0.62 | 0.82 | | | |
| Ca$^{2+}$ | 0.86 | ns | 0.83 | 0.69 | 0.82 | 0.61 | 0.76 | 0.71 | | |
| nss-SO$_4^{2-}$ | ns | ns | ns | ns | ns | ns | ns | ns | ns | ns |
| nss-Ca$^{2+}$ | 0.76 | ns | 0.75 | 0.77 | 0.68 | 0.56 | 0.66 | 0.56 | 0.96 | ns |

b)

| | Cl$^-$ | Br$^-$ | SO$_4^{2-}$ | NO$_3^-$ | Na$^+$ | NH$_4^+$ | K$^+$ | Mg$^{2+}$ | Ca$^{2+}$ | nss-SO$_4^{2-}$ |
|---|---|---|---|---|---|---|---|---|---|---|
| Br$^-$ | ns | | | | | | | | | |
| SO$_4^{2-}$ | 0.75 | 0.58 | | | | | | | | |
| NO$_3^-$ | ns | -0.48 | ns | | | | | | | |
| Na$^+$ | 0.95 | ns | 0.80 | ns | | | | | | |
| NH$_4^+$ | ns | ns | ns | ns | 0.47 | | | | | |
| K$^+$ | 0.83 | 0.46 | 0.73 | ns | 0.88 | 0.62 | | | | |
| Mg$^{2+}$ | 0.92 | ns | 0.78 | ns | 0.98 | 0.47 | 0.86 | | | |
| Ca$^{2+}$ | 0.85 | ns | 0.64 | 0.44 | 0.76 | ns | 0.62 | 0.70 | | |
| nss- SO$_4^{2-}$ | ns | ns | ns | ns | ns | ns | ns | ns | ns | ns |
| nss-Ca$^{2+}$ | 0.67 | ns | 0.45 | 0.56 | 0.54 | ns | ns | 0.47 | 0.91 | ns |