# Peer review of "Measurement report: Spatial variations in ionic chemistry and water stable isotopes in the snowpack on glaciers across Svalbard during the 2015-2016 snow accumulation season"

_Atmospheric Chemistry and Physics, 2020_

## Referee Comment (RC1) · Anonymous Referee #1 · 24 Sep 2020

Review on "Measurement report: Spatial variations in snowpack ionic chemistry and water stable isotopes across Svalbard" by Barbaro et al.

In this paper, the authors conducted snow observations on several glaciers in Svalbard in spring time, and showed spatial variations of the loads of chemical substances in snowpack and water stable isotopes. The spatial variations influenced by an unique location of Svalbard , which is in the boundary area of sea ice cover, are valuable data to evaluate transportation processes of chemical substances and water vapor surrounding Svalbard. Especially, a spatial variation of Br, which is reported by less common of previous research is very important, and suggests release processes of Br

[Figure]
* * *
Interactive
comment

from sea ice during sea ice formation. Moreover, the unification of observation methods of snowpack and chemical analyses conducted by several laboratories enhances the reliability of the data. For these reasons, I suggest that this manuscript is worthy of being a publication of ACP as a measurement report.

However, I believe that the data currently described in this manuscript are not enough. The authors also observed snow stratigraphy of snowpack and collected snow samples for chemical analyses from each snow layer according to the snow stratigraphy. Nevertheless, only total loads of chemical species in snowpack are reported in the current manuscript. I am strongly proposing that spatial variations of SWE-weighted mean concentrations as well as total load are at least reported in the main content of the manuscript and vertical profiles of stratigraphy of snow pack and concentration of chemical substances and water stable isotopes are reported in the main content of the manuscript or the supplementary materials. As the authors mentioned in the manuscript, the periods of snow accumulation are different from observation site to site. Therefore, load of chemical substances might be biased by the differences of the period of snow accumulation. Therefore, I believe that this measurement report should report both of spatial variations of loads and concentrations of chemical substances. Moreover, I can imagine that the spatial variations might be different from in autumn, winter, and early spring. Therefore, the vertical profiles of chemical substances indicating seasonal variations are also valuable.

I also have several concerns about the evaluations of the spatial variation in the results and the discussions.

Specific comments

Title: In this article, only seasonal snow (autumn, winter, and spring) is treated. The authors should titled in such a way that the seasonal snow is made explicit.

L90: Valence of Na ion is 1. Na+

L92: Main subject of Goto-Azuma et al. is influence of melt water on chemical profiles in snowpack. Winther et al. does not show chemical data. I believe that there are more appropriate references. For example, Issakson et al., 2001 showed the chemical data of ice core from Lomonosov fonna, and Matoba et al., 2002 showed chemical data of snowpack and ice core from Vestfonna.

L124. This protocol was used for the project of C2S3, and was not authorized by international organizations. Therefore, please change the description to avoid misunderstandings, and describe more details of the method.

Major ion analyses 2.2 Major ion analyses Please unify the items described in each analytical condition in each laboratory. Please added followings: Flow rate of IC at Hornsund and Venice. Information of the guard column in Uppsala Filtering method in Venice Authors used the terms "eluent" in Hornsund, and "mobile phase" in Uppsala and Venice. Please unify the term.

L158 Cation was determined in Uppsala. L174 In Barbaro et al., 2017, analytical methods of amino acid were described, but not of anion and cation. Please cite appropriate references.

2.2.4 Instrumental performance of each laboratory. I suggest that this part is described in the first chapter of 2.2.

3. Result As the authors mentioned, the periods of snow accumulation are different from observation site to site. Therefore, load of chemical substances might be biased by the differences of the period of snow accumulation. To evaluate the impacts of chemical substances on the environment of snowpack in Svalbard, the total loads of chemical substances are appropriate. On the other hand, to evaluate the transport process of chemical substances from the other regions to Svalbard, the concentrations of chemical substances are also appropriate.

L216 Please add the names of observation sites as "Hornsund area (HB and WB,

southern Spitsbergen)"

L219 same zone -> accumulation zone.

L223 Figure S1 is important data and should be in the main text, and not in the supplemental materials.

L216 "Hornsund area" L230 "Hornsund region", L234 "Hornsund glaciers" Please unify them.

L241 Figure S2 is important data and should be in the main text, and not in the supplemental materials. The spatial variation of water stable isotopes is very important.

L246 Please add the names of observation sites in "NW Spitsbergen (XXX, XXX).

L250-253 I could not catch what the authors would state.

4. Discussion Temporal variations of chemical substances are not subjects of this article. Therefore, the part L257-265 is not necessary for this article.

4.1 Why not specify the percentage of sea-salt derived components in all chemical species, and show the prominence of sea salt as an origin?

L281-288 For this discussion, nss-$Mg^{2+}$ should be used rather than total $Mg^{2+}$.

L297 Generally, $SO_x$ does not originate from biomass burning. Does the authors mean the transportation of the secondary aerosols of ammonium sulfate formed by $SO_x$ from coal combustion and ammonium from biomass burning?

L298-302 I could not catch what the authors would state.

L304-312I could not catch what the authors would state. At the ocean around Svalbard, algae bloom can occur in autumn. I am not sure that the autumn bloom can affect the snow chemistry in Svalbard. I just cite a reference (Andyna et al., 2014)

4.2 Chlorine depletion L349: Under the reaction of chlorine depletion, sea salt NaCl reacts with acid. Therefore, the acid should be described as $HNO_3$ and $H_2SO_4$ instead

of NO3- and SO42-.

L351 The weight ratio of Cl/Na is 1.8. 1.174 is the mole ratio of that. In this paper, weight units are used for loads of chemical substances. In Figure 3, Cl/Na ratio is 1.80. Please change the number of Cl/Na and confirm the calculation of Cl depletion rate.

Whitlow et al., (1992) indicated that Cl/Na in snowpack at the Summit of Greenland was higher than the ratio of seawater. According to this paper, Cl depletion occurs on aerosol particles and the Cl-depleted aerosol particles are scavenged during the transportation of air mass. Thus, Cl/Na of residue of air mass becomes higher values. On the other hand, Cl/Na in Svalbard is lower than seawater in this article. If the authors indicate the characteristics of snow in Svalbard based on the difference from Greenland, the worth of this articles should be higher.

4.3 Br enrichment Hara et al., 2017 showed that Br was concentrated on frost flowers formed on new sea ice before the end of polar night, and consistent with this article.

4.4 L394 "*" is used for PC calculation. "d=$\delta$2$\hat{I}\mathring{U}$ - 8 $\delta$18$\hat{I}$§" L395 d-excess is influenced by not only SST but also relative humidity and wind speed (Gat, 1996; Uemura et al., 2008)

4.5 L426-429 If the water vapor is long-range transported from lower latitude, isotopically heavier molecules should be removed from air mass during the transportation and delta value of 18O should be decreased. Moreover, it is not understandable that Na from sea splay is accumulated in air mass during the transportation of air mass from far south. Because, the amount of Na scavenged from the air mass is much larger than that input from sea spray on the ocean surface. I believe simply the reason for the correlation between d18O and Na concentration in the high latitude area is as below: At high latitude areas, air mass is transported from simply from low elevation site to high elevation site without any strong disturbance of atmospheric condition. In this case, isotopically heavier molecules and chemical substances from sea spley are scavenged from air mass gradually. Thus, d18O and Na concentration shows positive correlation.

References L532 Goto-Azuma et al., The name of the journal is "Publication of the International Association of Hydrological Sciences".

Figure 2 Caption. Please describe the name of observation site in the figure caption. as "HDF; Holtedahlfonna". KV should be KVG.

Table 1 What does the "air temperature" in Table 1 indicate? Please describe time and date.

References for this review. Andyna, M. et al., 2014, Geophys. Res. Lett., 41, doi 10.1002/2014GL061047. Hara et al., 2017, ACP, 17, 8577-8598. doi:10.5194/acp-17+8577-2017. Issakson et al., 2001, J. Glociol., 47, 335-345. doi: 10.3189/172756501781832313 Matoba et al., 2002, J. Geophys. Res., 107, D23, 4721.doi:10.1029/2002JD002205
* * *

---

## Referee Comment (RC2) · Anonymous Referee #2 · 9 Oct 2020

Review on "Measurement report: Spatial variation in snowpack ionic chemistry and water stable isotopes across Svalbard"

The paper presents the results of a snow chemistry survey carried out on several glaciers across Svalbard in the spring season. The authors analysed snow pit samples for their major ion loads and stable water isotope composition in order to understand the spatial distribution pattern of different chemical species in the snowpack across the archipelago. The comprehensiveness of the presented data set makes it valuable for improving our understanding of which role transport processes, deposition patterns and sea ice formation processes play for the chemical composition of the seasonal

snowpack on Svalbard. Therefore, I suggest the manuscript to be published in ACP as a measurement report.

However, according to the title of the paper setting focus on both ionic chemistry and stable water isotopes of the snowpack I am missing a more detailed description of the stable water isotope survey, in the methods, results and discussion section. From the introduction it does not really become clear to me for which reason stable water isotopes are analysed in this study. What is the major aim of the stable water isotope analyses and what is the connection to the ionic composition? What do you want to explain with the help of the stable water isotopes? I also propose to give a short overview of the current knowledge about stable water isotopes in snow on Svalbard in the introduction as it has been shortly done for the chemical impurities. Furthermore, in order to understand the distribution pattern of major ions and stable water isotopes on Svalbard I think it is highly necessary to give more details on the different glacier sites, i.e. about their similarities and differences regarding size, shape, exposition to wind, distance to the sea ice edge.

At the moment the authors provide values for ionic loads and stable water isotopes only as values summed up or averaged over the entire depth of the snowpack that has been sampled. Hence, I think that it is of high value for the manuscript if also the vertical distribution pattern of the ionic composition and of stable water isotopes is included as this can deliver additional information about seasonal differences in transport processes, moisture sources and accumulation. Therefore, I strongly recommend to add a chapter about the stratigraphic differences and similarities regarding the ionic load and the stable water isotope composition to the Results chapter of the manuscript.

When explaining the observed spatial differences in the ionic and isotopic composition of the snowpack at the different sites, I also strongly suggest to discuss the role of post-depositional processes, in particular the reallocation of snow, and thus reallocation of ions and isotopes due to wind drift in more detail. Katabatic winds blowing down glacier valleys might cause significant removal, redistribution and redeposition of snow and

thus might bias the ionic and isotopic composition of the snowpack. It is necessary to consider the aspect of wind drift in particular when trying to explain unexpected distribution patterns of ions and stable water isotopes (e.g. lack of altitudinal gradient in the stable water isotope composition).

Specific comments:

Title: The title should also indicate the seasonal aspect of the study.

Abstract: L46 Please write $\delta$D instead of $\delta$2H throughout the manuscript as this is the most common notation.

1. Introduction:

L81-82 It is unclear where the last part of the sentence ("... and has also experienced...") refers to – the west or the east of the archipelago. Please rephrase.

L 84-98 Please add some information about the state-of-the art of the spatial distribution of stable water isotopes in the snowpack on Svalbard.

L 100-110 Please state clearly what stable water isotopes are used for in this study, i.e. what is the objective connected with the analysis of stable water isotopes.

2. Methods:

L 118 Please give more details on the different glacier sites. How different are they regarding their size and shape? How are they oriented, i.e. what is there flow direction (N-S, W-E)? Are there differences in their exposition to wind (e.g. U-shaped valleys vs. plateaus)?

L122 Where does "these zones" refer to? Accumulation and ablation zones or the ELA? Please be precise here.

L128-129 Why do you use different sampling resolutions at the different sites? Treating all sites in the same way would increase the comparability of the data that you derive.

[Figure]

Please explain why you use this sampling strategy.

L 152 Delete "as recommended for this device and column" as this information is unnecessary.

L198-202 The description of the stable water analysis is very short. Please give more details here: How many injections did you use per sample? How was the raw data corrected (e.g. for memory effects, drift)? Which standards did you use?

3. Results:

It is worth to include a chapter about the vertical distribution of major ions and stable water isotopes in the snowpack, i.e. to add a chapter about the stratigraphy and the spatial differences (or similarities) between the different glacier sites.

L241 How did you calculate the SWE-weighted mean? Please add this information.

L249-251 Do you have any explanation for this unusual pattern? Due to the altitudinal effect one would expect the stable water isotope composition to become lighter with increasing height as you did observe on KGV, ALB, HDF and LF. What about reallocation of snow by wind drift? Are there differences in the exposition of the different glacier sites to wind drift? Is there any influence of the glacier size and shape on the snow accumulation pattern and thus potentially also on the distribution of the stable water isotope composition in the snow? What about post-depositional effects that might alter the isotopic composition of the snowpack, such as diffusion and sublimation? Please add some possible explanations to the discussion chapter 4.4.

4. Discussion:

L 259 What do you mean by "unpublished data"? Please specify.

L260 Finish the sentence after "firn cores" and start a new sentence "Our study . . .".

L 286 The sentence is incomplete. Please check.

L 315-319 I could not catch what you want to say here. Please rephrase and make two sentences.

L 336-338 What do you think might be the reason for this pattern? What about long-distance transport of pollutants to the site? Please discuss.

L395 The d excess is also influenced by the relative humidity in the moisture source region.

L 399-401 Please explain what the Kruskal-Wallis test is and what it is used for in general (statistically). Just from the values in Table 2, I cannot see a significant difference between the d excess values at AF and the other sites, e.g. a d excess of almost 17‰ also occurs at HDF3, values > 14‰ also occur at ALB1 and ALB2.

L 414 Actually, the d excess is the parameter that provides information about moisture source variations. $\delta$18O is rather related to the condensation temperature at the precipitation site. I suggest to also calculate the relation between log-transformed Na+ and the d excess and to compare the results with the relation between log-transformed Na+ and $\delta$18O.

L 430-433 What about reallocation of snow by wind drift? Depending on the exposition of the site it can significantly alter the ionic and isotopic composition of the snow after deposition.

Figure 1: Please add all glacier site abbreviations to the figure captions.

Technical corrections:

I suggest to use present tense throughout the manuscript whenever you are talking about results, in particular in the following paragraphs: L48-55, L216-253, L 281-288, L 290-302.

I also strongly recommend to make shorter sentences in order to increase the readability of the text, in particular in the following paragraphs: L58-59, L74-78, L105-108.

L106 Add comma after the bracket.

L124 Delete one dot behind "i.e." in the bracket.

L154 Delete the commas enclosing "as well as DPA".

L155 Add "the" before "solid phase".

L174 Delete the bracket before "Barbaro et al." and enclose 2017 by brackets.

L177 Use plural for "calibration".

L 178 Consider to use "delivered" instead of "gave".

L183 and L184 Add "all" before "cases".

L186 Add "the" before "bulk ionic load".

L214 Delete the comma after ELA.

L216 Replace "snow pits samples" by "snow-pit samples".

L 258-259 Delete the bracket before "Virkkunen et al." and enclose 2007 by brackets. The same for "Vega et al." (delete the bracket before and enclose the year in brackets).

L 263 Delete the bracket before "Spolaor et al." and enclose 2013 by brackets.

L 278 Add "the" before "snowpack".

L 284 Too many brackets re-write: (0.32; Millero etal., 2008; Figure 3).

L292 Write $Cl^-/SO_4^{2-}$ instead of $Cl^-:SO_4^{2-}$. Add "on" before "those".

L 300-301 Delete "which is a derived variable" as this has been explained before.

L 312 Add "an" before "underrepresentation".

L 359 Add a dot after the bracket and delete "and".

L 368 Delete the comma after "always".

L 369 Delete the bracket before "Jacobi et al." and enclose the year by brackets.

L 379 Delete the space after "this".

L 400 "was" needs to be shifted behind "d".

L 411 Delete the comma after "higher".

L 417 Replace "glaciers sites" by "glacier sites".

L 428 Replace "suggests" by "suggest".

Figure 2: Replace "glaciers zone" by "glacier zones".

Figure 3: L 705 Delete "the" before "spring".

Figure 5: L 726 Add "the" before "oxygen".

Table 1: Replace "Universit! de Franche-Comté" by "Université de Franche-Comté".

Table 2: Replace "compare to sea water" by "compared to sea water". Please also add the unit for $\delta$18O, $\delta$D and d excess in the table captions.

Table 4: Add a comma after Millero et al.

[Figure]

---

## Author Comment (AC1) · 3 Dec 2020

R: In this paper, the authors conducted snow observations on several glaciers in Svalbard in springtime and showed spatial variations of the loads of chemical substances in snowpack and water stable isotopes. The spatial variations influenced by an unique location of Svalbard, which is in the boundary area of sea ice cover, are valuable data to evaluate transportation processes of chemical substances and water vapour surrounding Svalbard. Especially, a spatial variation of Br, which is reported by less common of previous research is very important and suggests release processes of Br from sea ice during sea ice formation. Moreover, the unification of observation methods of snowpack and chemical analyses conducted by several laboratories enhances the reliability of the data. For these reasons, I suggest that this manuscript is worthy of being a publication of ACP as a measurement report. However, I believe that the data currently described in this manuscript are not enough. The authors also observed snow stratigraphy of snowpack and collected snow samples for chemical analyses from each snow layer according to the snow stratigraphy. Nevertheless, only total loads of chemical species in snowpack are reported in the current manuscript. I am strongly proposing that spatial variations of SWE-weighted mean concentrations as well as total load are at least reported in the main content of the manuscript and vertical profiles of stratigraphy of snow pack and concentration of chemical substances and water stable isotopes are reported in the main content of the manuscript or the supplementary materials.

A: We thank the reviewer for the useful comments. While we agree that going into the details of the layer resolution is of interest, we feel that it is out of the scope of this manuscript. In this paper, we present and discuss the spatial and altitudinal loads of different chemical species in the Svalbard Islands. For this reason, we omitted the description of the vertical snowpack profiles for each site and focused the discussion only on the total load, but have now included some figures that show the changes in snowpack density (Fig S4). In snow pits, it is rather difficult to associate a specific layer to a specific period and this makes the interpretation biased. In addition, warm events, which occurred during winter, could cause snow melting and percolation (another disturbance factor). This can be solved by including a snow pack model that is able to reproduce snow pack evolution over time and to associate a specific depth to a specific period. Moreover, snow modelling has to be associated to back trajectories analysis in order to identify possible sources of impurities in the snowpack. We proposed this manuscript as a measurement report since we wanted to have a dedicated study focusing on the regional differences in loading. The layer by layer investigation of the Svalbard glaciers, using the data collected in the accumulation areas, will be the aim of the next paper where all the associate variables will be considered. However, if the referee would like to have our dataset to check some calculations, we can send it

privately.

R: As the authors mentioned in the manuscript, the periods of snow accumulation are different from observation site to site. Therefore, load of chemical substances might be biased by the differences of the period of snow accumulation. Therefore, I believe that this measurement report should report both of spatial variations of loads and concentrations of chemical substances.

A: We agree with the referee that it is useful to add both loads and concentrations. In the previous version of the manuscript, we attempted to address this by calculating the correlation coefficients both for loads and for concentrations. However, as suggested by referee, we added a new table with volume-weighted mean concentrations of major ions in each snow pit (calculated as the sum of concentrations in all layers divided by the total SWE of the snow pit) in this revised version to improve clarity.

R: Moreover, I can imagine that the spatial variations might be different from in autumn, winter, and early spring. Therefore, the vertical profiles of chemical substances indicating seasonal variations are also valuable. I also have several concerns about the evaluations of the spatial variation in the results and the discussions.

A: We are agree with the referee, but the identification of the seasonal variation requires snow pack modelling for each site studied and all evaluations listed in our previous answer. The aim of the paper is to understand the regional differences in the annual snow pack and not investigate the differences for each snow layer and the variables that could affect the snowpack stratigraphy. We present the paper as a measurements report since we aim to have an initial large-scale evaluation of the main processes and then specifically address the snowpack stratigraphy. We feel that this initial analysis is a critical first step before proceeding with the detailed evaluation of each layer. In addition, we feel that it is important to make our measurements available to the larger research community.

R: Specific comments R: Title: In this article, only seasonal snow (autumn, winter, and

spring) is treated. The authors should titled in such a way that the seasonal snow is made explicit.

A: We modified the title as follows: "Measurement report: Spatial variations in seasonal snowpack ionic chemistry and water stable isotopes across Svalbard"

R: L90: Valence of Na ion is 1. Na+

A: Sorry for this terrible mistake for a chemist.

R: L92: Main subject of Goto-Azuma et al. is influenced of melt water on chemical profiles in snowpack. Winther et al. does not show chemical data. I believe that there are more appropriate references. For example, Issakson et al., 2001 showed the chemical data of ice core from Lomonosovfonna, and Matoba et al., 2002 showed chemical data of snowpack and ice core from Vestfonna.

A: Thanks for the suggestion. We modified as suggested by the referee, removing Goto Azuma and Winther and adding Issakson et al. and Matoba et al.

R: L124. This protocol was used for the project of C2S3 and was not authorized by international organizations. Therefore, please change the description to avoid misunderstandings and describe more details of the method.

A: As suggested by referee 1, the protocol is not authorized by international organizations, so we removed "standardized" from the sentence. In the manuscript, we included the description of the main points of this common protocol.

R: Major ion analyses 2.2 Major ion analyses. Please unify the items described in each analytical condition in each laboratory. Please added followings: Flow rate of IC at Hornsund and Venice. Information of the guard column in Uppsala Filtering method in Venice Authors used the terms "eluent" in Hornsund, and "mobile phase" in Uppsala and Venice. Please unify the term.

A: We unified the analytical conditions description in each lab, adding the flow rate at

Hornsund and several method details at Venice lab (gradient condition, flow rate). We used "mobile phase" in each description.

R: L158 Cation was determined in Uppsala.

A: Yes, this is the description of cation determination, as suggested by the column and mobile phase. Sorry for the mistake.

R: L174 In Barbaro et al., 2017, analytical methods of amino acid were described, but not of anion and cation. Please cite appropriate references.

A: We agree with the referee, this reference is wrong. We modified the reference to the correct Barbaro et al. 2017. (Barbaro et al. Particle size distribution of inorganic and organic ions in coastal and inland Antarctic aerosol. Environmental Science and Pollution Research, 2017, 24.3: 2724-2733.)

R: 2.2.4 Instrumental performance of each laboratory. I suggest that this part is described in the first chapter of 2.2.

A: We feel that introducing the instrumental analysis details and then describing the instrumental performance of each lab provides more clarity.

R: 3. Result As the authors mentioned, the periods of snow accumulation are different from observation site to site. Therefore, load of chemical substances might be biased by the differences of the period of snow accumulation. To evaluate the impacts of chemical substances on the environment of snowpack in Svalbard, the total loads of chemical substances are appropriate. On the other hand, to evaluate the transport process of chemical substances from the other regions to Svalbard, the concentrations of chemical substances are also appropriate.

A: In the submitted manuscript, we have already considered the data both as load and as concentrations ("we computed Spearman rank correlations between total ionic loads (load), as well as between volume-weighted mean ionic concentrations (conc)"). On the other hand, we agree with the referee that we have to add a new table (Table

3) with the raw data of the volume-weighted mean concentrations of each ion. We add these sentences in the manuscript: "On the other hand, to evaluate the transport processes of chemical species from the other regions to Svalbard, we evaluate the volume-weighted mean concentrations of major ions in each snow pit. These values are calculated as the total ionic load of each snow pit divided by its total SWE (snow water equivalent) (Table 3)."

R: L216 Please add the names of observation sites as "Hornsund area (HB and WB, southern Spitsbergen)"

A: Thanks for the suggestion, we added HB and WB inside the brackets.

R: L219 same zone -> accumulation zone.

A: We substituted "same" with "accumulation".

R: L223 Figure S1 is important data and should be in the main text, and not in the supplemental materials.

A: As suggested by referee 1, we moved figure S1 (now Figure 3) into the main manuscript and we corrected all following figure numbers.

R: L216 "Hornsund area" L230 "Hornsund region", L234 "Hornsund glaciers" Please unify them.

A: We now use always "Hornsund area" instead of "Hornsund region" but we maintained "Hornsund glaciers" because this refers to "snowpack on Hornsund glaciers".

R: L241 Figure S2 is important data and should be in the main text, and not in the supplemental materials. The spatial variation of water stable isotopes is very important.

A: As suggested by referee 1, we moved the Figure S2 (now Figure 4) in the main manuscript and we corrected all following figure numbers.

R: L246 Please add the names of observation sites in "NW Spitsbergen (XXX, XXX).

A: As suggested by the referee, we added "(KVG, ALB and HDF)".

R: L250-253 I could not catch what the authors would state.

A: To clarify the statement, we simplified as follows: "On the other hand, AF, WB, and HB there was no statistical difference between the mean $\delta 18O$ and $\delta 2H$ values (Figure 4). "

R: 4. Discussion. Temporal variations of chemical substances are not subjects of this article. Therefore, the part L257-265 is not necessary for this article.

A: We included this part because we want to compare our data with previous measurements reported in the Svalbard glaciers. The state of the art is important to define if our data is comparable with previous studies (i.e., representative for more than just the 2015/16 accumulation season).

R: 4.1 Why not specify the percentage of sea-salt derived components in all chemical species, and show the prominence of sea salt as an origin?

A: In Table 3, we included the volume-weighted mean concentrations of major ions in each snow pit (calculated as the sum of concentrations in all layers divided by the total SWE of the snow pit). Here we included also nss-SO42-, nss- K+, nss- Mg2+, nss-Ca2+. The discussion about these fractions is reported in the main manuscript.

L281-288 For this discussion, nss-Mg2+ should be used rather than total Mg2+.

A: As shown in the table 3, the concentrations of nss-Mg2+ are frequently negative. Therefore, to avoid introducing extra uncertainty, we avoided performing a correlation with Ca2+ and nss-Mg2+.

L297 Generally, SOx does not originate from biomass burning. Does the authors mean the transportation of the secondary aerosols of ammonium sulfate formed by SOx from coal combustion and ammonium from biomass burning?

A: To clarify the concept, we modified the sentence as follows: "Another plausible

source of nss-SO42- deposition in Svalbard is long-range atmospheric transport of secondary aerosols containing SO42-, such as ammonium sulfate. This sulphate can be formed by SOx emitted from coal combustion throughout the winter and biomass burning in the spring (Barrie, 1986; Law and Stohl, 2007; Nawrot et al., 2016)"

R: L298-302 I could not catch what the authors would state.

A: We clarified the concept as follows: "However, we need to caution that in the southern region of Svalbard, the estimation of ss-SO42- is subject to higher uncertainty because of the higher amount of Na+ in the atmospheric deposition there."

L304-312I could not catch what the authors would state. At the ocean around Svalbard, algae bloom can occur in autumn. I am not sure that the autumn bloom can affect the snow chemistry in Svalbard. I just cite a reference (Andyna et al., 2014)

A: We refer to the algal bloom that can occur in May or late spring. In our recent publication (Spolaor, Andrea et al. "Source, timing and dynamics of ionic species mobility in the Svalbard annual snowpack." Science of The Total Environment 751 (2020): 141640. ), the dynamics of ionic species are investigated and we found an increase in surface concentrations of sulphate and MSA during May. The sampling performed in April can underestimate the impact of nss-SO4 coming from biogenic sources.

R: 4.2 Chlorine depletion L349: Under the reaction of chlorine depletion, sea salt NaCl reacts with acid. Therefore, the acid should be described as HNO3 and H2SO4 instead of NO3- and SO42-.

A: We completely agree with the referee because the reactions are the following: HNO3 (aq) +NaCl (aq,s) → NaNO3 (aq,s) + HCl (g) H2SO4 (aq) +2NaCl (aq,s) →Na2SO4 (aq,s) +2 HCl (g) We modified it as suggested by the referee.

R: L351 The weight ratio of Cl/Na is 1.8. 1.174 is the mole ratio of that. In this paper, weight units are used for loads of chemical substances. In Figure 3, Cl/Na ratio is 1.80. Please change the number of Cl/Na and confirm the calculation of Cl depletion rate.

A: Figure 3 (now 5) reports the weight ratio and 1.8 is corrected as also confirmed by referee. The values of Cl depletion (%) were calculated with the formula reported in the manuscript and we used the equivalent concentration to calculate the Cl depl. Therefore, we can confirm that the values of Cl depl are correct.

R: Whitlow et al., (1992) indicated that Cl/Na in snowpack at the Summit of Greenland was higher than the ratio of seawater. According to this paper, Cl depletion occurs on aerosol particles and the Cl-depleted aerosol particles are scavenged during the transportation of air mass. Thus, Cl/Na of residue of air mass becomes higher values. On the other hand, Cl/Na in Svalbard is lower than seawater in this article. If the authors indicate the characteristics of snow in Svalbard based on the difference from Greenland, the worth of this article should be higher.

A: Thanks for the suggestion; we introduced a comparison between Cl/Na in the snowpack of Svalbard and Greenland. We introduced these sentences in section 4.2: "Withlow et al. (1992) found an opposite situation in the snowpack of Greenland, indicating Cl-/Na+ values higher than the ratio of seawater. This Cl- enrichment relative to the Cl-/Na+ ratio in sea water may reflect Cl derived from anthropogenic sources as well from gas phase chlorine transportation and deposition in central Greenland."

R: 4.3 Br enrichment Hara et al., 2017 showed that Br was concentrated on frost flowers formed on new sea ice before the end of polar night, and consistent with this article.

A: We thank the referee for finding similarities between our results and Hara et al. 2017. The Br enrichment (or depletion) in the snowpack is not only due to physical atmospheric sea spray transport (or Br from frost flowers on sea ice) but also to the atmospheric chemical processes that can alter its signature The aim of the paper is broader than Br chemistry, which is why we chose not to go into the details of Br atmospheric processes, but their description is reported in the references cited. Bromine activation requires higher amounts of salts to sustain the "explosion" and the sea ice surface (mainly first-year sea ice) is the perfect substrate to start the so-called bromine

explosion. However, frost flowers might not be the direct source of extra bromine, since no fractions of Br compared to Na have been determined and suggested. On the other hand, frost flowers could be a source of gas phase bromine, although not the only one, since the snow over sea ice can act as a source. Hara and co-authors did their study in Antarctica where sea ice is mainly as first-year sea ice. The sea ice around Svalbard is also mainly first-year ice, but the sea ice drift occurring in the Arctic basin could replace the first-year sea ice with multi-year sea ice, making the condition different. This process is rather evident in the Fram Strait where a large amount of sea ice is expulsed from the Arctic basin.

R:4.4 L394 "*" is used for PC calculation.

A: We removed "*" and we added "Å̊" to indicate the multiplication.

R: L395 d-excess is influenced by not only SST but also relative humidity and wind speed (Gat, 1996; Uemura et al.,2008)

A: We modified the sentence as follows: "Deuterium excess (d = $\delta$D-(8Å̊$\delta$18O)) is mainly influenced by the source region of the precipitating moisture and in particular by the sea surface temperature, but also relative humidity and wind speed."

R: 4.5 L426-429 If the water vapor is long-range transported from lower latitude, isotopically heavier molecules should be removed from air mass during the transportation and delta value of 18O should be decreased. Moreover, it is not understandable that Na from sea splay is accumulated in air mass during the transportation of air mass from far south. Because, the amount of Na scavenged from the air mass is much larger than that input from sea spray on the ocean surface. I believe simply the reason for the correlation between d18O and Na concentration in the high latitude area is as below: At high latitude areas, air mass is transported from simply from low elevation site to high elevation site without any strong disturbance of atmospheric condition. In this case, isotopically heavier molecules and chemical substances from sea spray are scavenged from air mass gradually. Thus, d18O and Na concentration shows positive

correlation.

A: We thank the referee for this comment. We agree with some aspects and have modified the text in the manuscript accordingly. We think that the explanation of the correlation between $\delta 18O$ and Na is not straightforward and a few hypotheses should be considered. However, we do not think that a simple atmospheric distillation is the main reason, because otherwise the same correlation should be found at any elevation, assuming a similar $\delta 18O$ and Na fractionation. We agree that the sodium might not be collected during air mass transport from lower latitudes and we removed this hypothesis as suggested by the referee. The text has been modified as follows: "The increase in strength and significance of the log(Naload)-$\delta 18O$ correlation with altitude might be explained by different contributions of locally emitted ssNa+, relative to those of Na+ from more distant sources. Sites located at lower altitudes are proportionally more affected by local sea spray deposition, with or without snowfall. Conversely, sites at higher elevations likely receive a larger share of their ionic load from more distant sources, and by wet deposition through snowfall. At the four sites (KVG, AF, LF, and HDF) where the log(Naload)-$\delta 18O$ correlation was significant, increases in $\delta 18O$ in snow layers were often associated with higher Na+ concentrations. It is rather difficult to propose a precise explanation for this association. However, we would suggest that the isotopically heavier (less negative) $\delta 18O$ values suggest that the co-registered Na+ enhancements were associated with precipitation of relatively warm air, probably advected from lower latitudes. The snowfall associated with a warm event is able to wet scavenge the sea spray aerosol present in the atmosphere. On the contrary, when the cold air masses (Arctic type) dominated, the snowfall events were relatively limited due to the poor air humidity causing a lower efficiency of wet scavenging. This resulted in lower $\delta 18O$ and (likely) Na sodium loads, suggesting that wet deposition dominated the chemical load of the snowpack. Although this process is should occur also at lower elevation sites, the local emission and associated dry deposition are likely more important than wet deposition; more frequent melt-refreeze episodes at lower elevations would also mask the relationship proposed (as suggested by the vertical profiles of

stratigraphy reported in Figure S4). Another possible explanation is that in the Arctic, air masses are transported from low to high elevation sites without any strong disturbance of the atmospheric conditions. In this case, isotopically heavier molecules and sea spray particles are gradually scavenged from the air masses. If this was the main process, we should find the correlation across all studied sites, assuming that Na+ scavenged at a similar rate as that of isotopic fractionation. Since this has not been found, we propose that the correlation at higher elevation cannot be explained by atmospheric distillation alone. The possibility that the correlation is due to different sources of air masses seems unsupported due to the absence of correlation between d-excess and sodium."

R: L532 Goto-Azuma et al., The name of the journal is "Publication of the International Association of Hydrological Sciences".

A: We modified the name of the journal as suggested by referee.

R: Figure 2 Caption. Please describe the name of observation site in the figure caption.as "HDF; Holtedahlfonna". KV should be KVG.

A: Done

Table 1 What does the "air temperature" in Table 1 indicate? Please describe time and date.

A: As suggested by the referee, we clarify the parameter in the caption as follows: "The air temperature was measured with a digital thermometer when the operators started to dig the snow pits."

References for this review. Andyna, M. et al., 2014, Geophys.Res.Lett.,41, doi 10.1002/2014GL061047. Hara et al., 2017, ACP, 17, 8577-8598.doi:10.5194/acp-17+8577-2017. Issakson et al., 2001, J. Glociol., 47, 335-345. doi:10.3189/172756501781832313 Matoba et al., 2002, J. Geophys. Res., 107, D23,4721.doi:10.1029/2002JD002205

Please also note the supplement to this comment:
https://acp.copernicus.org/preprints/acp-2020-740/acp-2020-740-AC1-supplement.pdf

───────────────────────────────

---

## Author Comment (AC2) · 3 Dec 2020

R: The paper presents the results of a snow chemistry survey carried out on several glaciers across Svalbard in the spring season. The authors analysed snow pit samples for their major ion loads and stable water isotope composition in order to understand the spatial distribution pattern of different chemical species in the snowpack across the archipelago. The comprehensiveness of the presented data set makes it valuable for improving our understanding of which role transport processes, deposition patterns and sea ice formation processes play for the chemical composition of the seasonal snowpack on Svalbard. Therefore, I suggest the manuscript to be published in ACP

as a measurement report. However, according to the title of the paper setting focus on both ionic chemistry and stable water isotopes of the snowpack I am missing a more detailed description of the stable water isotope survey, in the methods, results and discussion section. From the introduction it does not really become clear to me for which reason stable water isotopes are analysed in this study. What is the major aim of the stable water isotope analyses and what is the connection to the ionic composition? What do you want to explain with the help of the stable water isotopes? I also propose to give a short overview of the current knowledge about stable water isotopes in snow on Svalbard in the introduction as it has been shortly done for the chemical impurities.

A: Thanks to the referee for the comments. We improved the stable isotope description to clarify the aim of the use of these ratios in the manuscript and to describe the state-of-the-art of these measurements in the Svalbard snowpack. Please see the specific comments where we reported all modifications.

R: Furthermore, in order to understand the distribution pattern of major ions and stable water isotopes on Svalbard I think it is highly necessary to give more details on the different glacier sites, i.e. about their similarities and differences regarding size, shape, exposition to wind, distance to the sea ice edge.

A: As proposed by the referee, we included a new paragraph where we described each glacier monitored.

R: At the moment the authors provide values for ionic loads and stable water isotopes only as values summed up or averaged over the entire depth of the snowpack that has been sampled. Hence, I think that it is of high value for the manuscript if also the vertical distribution pattern of the ionic composition and of stable water isotopes is included as this can deliver additional information about seasonal differences in transport processes, moisture sources and accumulation. Therefore, I strongly recommend to add a chapter about the stratigraphic differences and similarities regarding the ionic load and the stable water isotope composition to the Results chapter of the manuscript.

[Figure]

When explaining the observed spatial differences in the ionic and isotopic composition of the snowpack at the different sites, I also strongly suggest to discuss the role of post-depositional processes, in particular the reallocation of snow, and thus reallocation of ions and isotopes due to wind drift in more detail. Katabatic winds blowing down glacier valleys might cause significant removal, redistribution and redeposition of snow and thus might bias the ionic and isotopic composition of the snowpack. It is necessary to consider the aspect of wind drift in particular when trying to explain unexpected distribution patterns of ions and stable water isotopes (e.g. lack of altitudinal gradient in the stable water isotope composition).

A: We thank referee 2 for this comment and it is also a point raised by referee 1. We beg to differ at this point: we think that presenting the data at layer resolution will exceed significantly the intended scope of this manuscript (there could be a whole paper written on that topic, which is in preparation). In this paper, we present and discuss the spatial and altitudinal differences in the loads of several chemical species. The evaluation of the layer composition requires extensive scientific discussion and modelling work. In snow pits, it is rather difficult to associate a specific layer with a specific period and this makes the interpretation biased. This can be solved by including snow pack modelling to reproduce temporally the snow pack evolution and so to link specific depths to specific periods. To identify possible sources of impurities in the snowpack, such studies should be complemented by air mass back trajectory analysis. In addition, warm events occurring during winter could cause snow melting and percolation (another disturbance factor). A detailed insight into all these processes was not the aim of this manuscript, on the contrary, we mainly focused on characterising and identifying regional differences of chemical species across Svalbard. For this reason, we omit the description of the vertical snowpack profiles for each site and we focus the discussion only on the total load. For fair assessment of our work (e.g., to check the calculations), we are happy to make the layer-resolution dataset available for the reviewer, but we do not intend to publish it in this Measurement report.

Specific comments: R: Title: The title should also indicate the seasonal aspect of the study.

A: We modified the title as follows: "Measurement report: Spatial variations in seasonal snowpack ionic chemistry and water stable isotopes across Svalbard"

Abstract: L46 Please write $\delta D$ instead of $\delta 2H$ throughout the manuscript as this is the most common notation.

A: We agree with the reviewer that different notations can be used and that most journals accepted both versions. $\delta D$ is historical and a new "Unit" was proposed in the SI system about 10 years ago. This unit has not been used widely, but as compromise today most people use $\delta 2H$ and not $\delta D$. ($\delta D$ is not included in the SI-system). We based the use of $\delta 2H$ on the following references. References Dunn, Philip & Carter, J.F.. (2018). Good Practice Guide for Isotope Ratio Mass Spectrometry Second Edition 2018. Pag 8 Brand, Willi A., et al. "Assessment of international reference materials for isotope-ratio analysis (IUPAC Technical Report)." Pure and Applied Chemistry 86.3 (2014): 425-467. Meier-Augenstein, Wolfram, and Arndt Schimmelmann. "A guide for proper utilisation of stable isotope reference materials." Isotopes in environmental and health studies 55.2 (2019): 113-128.

1. Introduction: R: L81-82 It is unclear where the last part of the sentence ("...and has also experienced...") refers to – the west or the east of the archipelago. Please rephrase.

A: We modified the sentence as follows: "the west exhibits higher temperatures and precipitation, while the east is less humid and cooler, and has also experienced a stronger warming trend since 1957."

R: L 84-98 Please add some information about the state-of-the art of the spatial distribution of stable water isotopes in the snowpack on Svalbard

A: We added a small paragraph in the introduction to define the state-of the art of stable

isotopes in Svalbard.

R: L 100-110 Please state clearly what stable water isotopes are used for in this study, i.e.what is the objective connected with the analysis of stable water isotopes.

A: To clarify the aim of stable water isotope determination we added this sentence: "Stable isotope ratios were used as supporting data to define the accumulation seasonality in snowpack, and to identify the moisture sources that feed snowfall, thereby providing clues to the predominant air transport pathways to the snow pit sites (Gat et al., 2001)."

2. Methods: L 118 Please give more details on the different glacier sites. How different are they regarding their size and shape? How are they oriented, i.e. what is there flow direction (N-S, W-E)? Are there differences in their exposition to wind (e.g. U-shaped valleys vs. plateaus)?

As suggested by referee, we added a new part with the description of each glacier.

L122 Where does "these zones" refer to? Accumulation and ablation zones or the ELA? Please be precise here.

A: As suggested by referee, we added "(accumulation, ablation, and ELA)".

R: L128-129 Why do you use different sampling resolutions at the different sites? Treating all sites in the same way would increase the comparability of the data that you derive. Please explain why you use this sampling strategy.

A: In this work, we used the protocol published by Gallet al al. (2018). The authors clearly reported the advantage of sampling per discrete layers. This type of sampling allows to link a snow layer (and its properties) to a specific climate event (i.e. precipitation or surface melt). The investigation of ionic composition of snow helps to identify the atmospheric sources of impurities deposited in the snowpack during a specific snow accumulation period. Moreover, sampling by discrete layer makes it possible to correlate intervals of snow accumulation between separate snow pits at different altitudes,

as reported in this paper when we compared the three different areas in the same glacier (ablation, ELA and accumulation). For the purpose of total load calculations, this procedure is also better because it treats as entities layers which are relatively homogeneous, allowing a more appropriate inclusion of ice layers into both density and chemistry measurements. Contrastingly, the sampling by fixed depth increments is a more simple and easy procedure to use by multiple teams because it does not require a description of the snowpack stratigraphy by an expert operator. To improve the manuscript we added these sentences: "This type of sampling facilitates linking a snow layer (and its properties) to a specific weather event (i.e., precipitation or surface melt). Moreover, sampling by discrete layers makes it possible to correlate the intervals of snow accumulation between separate snow pits at different altitudes, as reported in this paper when we compared the three different areas in the same glacier (ablation, ELA and accumulation). It is also more accurate for chemical load calculations where ice layers occur in snow pits."

R: L 152 Delete "as recommended for this device and column" as this information is un-necessary.

A: As suggested by referee, we removed this part of sentence.

R: L198-202. The description of the stable water analysis is very short. Please give more details here: How many injections did you use per sample? How was the raw data corrected (e.g. for memory effects, drift)? Which standards did you use?

A: We improved the description of stable water isotope analysis as follows: "The determination of stable isotope ratios of O and H was performed at Tallinn University of Technology (Estonia). The isotopic ratios were determined by laser spectroscopy, using a Picarro model L2120-i water isotope analyzer (Picarro Inc., Sunnyvale, USA), which allows for the simultaneous determinations of 18O/16O and 2H/1H in H2O with a high-precision AO211 vaporizer. Results are reported in the standard delta notation as $\delta$18O and $\delta$2H relative to Vienna Standard Mean Ocean Water (VSMOW). Reproducibility was ±0.1‰ for $\delta$18O and ±1‰ for $\delta$2H, respectively. 7 injections were carried out for each sample, but only the last 4 injections (4 to 7) were used for calculations to minimize the memory effect. Laboratory standards TLN-A2 (-10.15; -77.5) and TLN-B2 (-21.95; -162.5) were regularly calibrated against international V-SMOW, GNIP and V-SLAP standards. Standards (TLN-A2, TLN-B, and TLN-D4) were measured at the beginning, in the middle, and at the end of each set of measurements (54 bottles). Additionally, every 7 samples, the laboratory standard TLN-D4 (-17.5; -133.0) was measured and used for drift correction if needed."

R: 3. Results: It is worth to include a chapter about the vertical distribution of major ions and stable water isotopes in the snowpack, i.e. to add a chapter about the stratigraphy and the spatial differences (or similarities) between the different glacier sites.

A: Please check the answer in the general comments. We think this would need more than a chapter – rather a separate article.

R: L 241 How did you calculate the SWE-weighted mean? Please add this information.

A: As suggested by referee, we clarified this calculation: "The SWE-weighted mean $\delta$18O and $\delta$2H are calculated using the formula SWE-$\delta$ = $\sum(\delta$i x SWEi)/SWEt where $\delta$i areis the con the $\delta$ values of each layer, SWEi are SWE of each layer and SWEt is the SWE of the entire snow pit."

R: L249-251 Do you have any explanation for this unusual pattern? Due to the altitudinal effect one would expect the stable water isotope composition to become lighter with increasing height as you did observe on KGV, ALB, HDF and LF. What about reallocation of snow by wind drift? Are there differences in the exposition of the different glacier sites to wind drift? Is there any influence of the glacier size and shape on the snow accumulation pattern and thus potentially also on the distribution of the stable water isotope composition in the snow? What about post-depositional effects that might alter the isotopic composition of the snowpack, such as diffusion and sublimation? Please add some possible explanations to the discussion chapter 4.4.

A: The post- depositional processes could influence the water stable isotope signal, but there is still lack of knowledge regarding the effect of post-depositional processes on the preserved signal. For example, post-depositional processes could equilibrate the snow water stable isotope value with the average value characterising the atmosphere above the site of sampling. Defining the role of the post-depositional processes is rather difficult and giving an estimation at this point would be speculative. Certainly, wind redistribution could have an effect as well but the snow redistribution might be confined to the area of snow deposition (within < 1km). Generally the occurrence of strong wind produces the so called wind crust, an extremely hard snow layer difficult for the wind to lift. All the snow pits have been collected from the glacier central line in order to minimise the side accumulation effect due to orography.

4. Discussion: R: L 259 What do you mean by "unpublished data"? Please specify.

A: As suggested by referee we clarified this aspect: ". In table 4, we report also unpublished data of samples collected in 2009-2011 by C. Vega, obtained using the same methods reported in section 2.2."

R: L260 Finish the sentence after "firn cores" and start a new sentence "Our study...".

A: As suggested by referee, we split the sentence.

R: L 286 The sentence is incomplete. Please check

A: Sorry for the mistake. We completed the sentence as follows: "which are abundant in Svalbard (Dallmann, 1999)."

L 315-319 I could not catch what you want to say here. Please rephrase and make two sentences.

A: We clarified the sentence as suggested by referee.

R: L 336-338 What do you think might be the reason for this pattern? What about long-distance transport of pollutants to the site? Please discuss.

A: The explanation of this pattern was reported in the following sentences: "The highest correlation coefficient for NO3-, both in terms of concentrations and loads, was found with nss-Ca2+. This would support both the formation of calcium nitrate in the atmosphere (Gibson et al., 2006) or post-depositional processes removing the NO3- from layers poor in Ca2+, since calcium has been hypothesised to stabilise the nitrate in snowpack against post-depositional losses (Kekonen et al., 2017)."

L395 The d excess is also influenced by the relative humidity in the moisture source region.

A: As suggested by both referees, we specified this aspect as follows: "Deuterium excess (d = $\delta$2H-(8·$\delta$18O)) is mainly influenced by the source region of the precipitating moisture and in particular by the sea surface temperature, but also relative humidity and wind speed (Gat, 1996; Uemura et al., 2008). In addition, d is also influenced by the temperature gradient between the moisture source and precipitation area (Johnsen et al., 1989)."

L 399-401 Please explain what the Kruskal-Wallis test is and what it is used for in general (statistically). Just from the values in Table 2, I cannot see a significant difference between the d excess values at AF and the other sites, e.g. a d excess of almost 17‰ also occurs at HDF3, values > 14‰ also occur at ALB1 and ALB2.

A: To clarify this concept, we modified the sentence as follows: "more detailed analysis of d by latitude showed that only significantly different values were obtained in snow pits sampledonly beyond 79.2 °N, i.e., in Austfonna snow pits. This is confirmed by (the Kruskal-Wallis test, , i.e. rank-based ANOVA, calculated with two groups of d values divided by the latitude threshold 79.2°N; z = 4.23, p < 0.04; in fact, drawing the latitude threshold anywhere between 78.7 and 79.7 °N resulted in a statistically significant difference with p < 0.05)."

L 414. Actually, the d excess is the parameter that provides information about moisture source variations. $\delta$18O is rather related to the condensation temperature at the

precipitation site. I suggest to also calculate the relation between log-transformed Na+ and the d excess and to compare the results with the relation between log-transformed Na+ and $\delta$18O.

A: Thanks to referee for this comment, similar to the observation of referee 1. We agree that the increased sodium concentration might not be related to the longer atmospheric pathway above the ocean surface but might be also linked with the wet scavenging occurring during a snowfall event. The correlation determined is not due to the source but instead the occurrence of a snowfall able to clean up the atmosphere and normally associated to relatively warmer and wetter air masses as compared to the Arctic type. This hypothesis is also supported by the lack of correlation between d-excess and sodium in our dataset. The d-excess in this case is not useful since, as the referee states, it is an indicator of air mass sources. We modified the main manuscript as follows to better explain the correlation between log(Naload)-$\delta$18O: "At the four sites (KVG, AF, LF, and HDF) where the log(Naload)-$\delta$18O correlation was significant, increases in $\delta$18O in snow layers were often associated with higher Na+ concentrations. It is rather difficult to propose a precise explanation for this association. However, we would suggest that the isotopically heavier (less negative) $\delta$18O values suggest that the co-registered Na+ enhancements were associated with precipitation of relatively warm air, probably advected from lower latitudes. The snowfall associated with a warm event is able to wet scavenge the sea spray aerosol present in the atmosphere. On the contrary, when the cold air masses (Arctic type) dominated, the snowfall events were relatively limited due to the poor air humidity causing a lower efficiency of wet scavenging. This resulted in lower $\delta$18O and (likely) Na sodium loads, suggesting that wet deposition dominated the chemical load of the snowpack. Although this process is should occur also at lower elevation sites, the local emission and associated dry deposition are likely more important than wet deposition; more frequent melt-refreeze episodes at lower elevations would also mask the relationship proposed (as suggested by the vertical profiles of stratigraphy reported in Figure S4). Another possible explanation is that in the Arctic, air masses are transported from low to high elevation sites

without any strong disturbance of the atmospheric conditions. In this case, isotopically heavier molecules and sea spray particles are gradually scavenged from the air masses. If this was the main process, we should find the correlation across all studied sites, assuming that Na+ scavenged at a similar rate as that of isotopic fractionation. Since this has not been found, we propose that the correlation at higher elevation cannot be explained by atmospheric distillation alone. The possibility that the correlation is due to different sources of air masses seems unsupported due to the absence of correlation between d-excess and sodium."

R: L 430-433 What about reallocation of snow by wind drift? Depending on the exposition of the site it can significantly alter the ionic and isotopic composition of the snow after deposition.

A: As reported in the previous answer, wind drift is a very important topic that this paper cannot appropriately solve. We dug the snow pits along the central line in order to minimize the orography interferences. However, the effect of wind should be limited due to the formation of wind crust. We admit that the reallocation of chemical species in the snowpack is a very interesting topic but we would need another type of experiment to evaluate it. Here, the aim is to define the spatial distribution of chemical loads.

R: Figure 1: Please add all glacier site abbreviations to the figure captions.

A: We added this sentence: "Abbreviation: KVG = Kongsvegen, HDF = Holtedahlfonna, AF= Austfonna, ALB = Austre Lovénbreen, LF = Lomonosovfonna, HB = Hansbreen, WB = Werenskiöldbreen."

Technical corrections: I suggest to use present tense throughout the manuscript whenever you are talking about results, in particular in the following paragraphs: L48-55, L216-253, L 281-288,L 290-302.I also strongly recommend to make shorter sentences in order to increase the readability of the text, in particular in the following paragraphs: L58-59, L74-78, L105-108.

A: We modified the entire manuscript as suggested by referee. R: L106 Add comma after the bracket.

A: Done

R: L124 Delete one dot behind "i.e." in the bracket.

A: We removed the comma after i.e.

R: L154 Delete the commas enclosing "as well as DPA".

A: Done

R: L155 Add "the" before "solid phase".

A: We added "the"

R: L174 Delete the bracket before "Barbaro et al." and enclose 2017 by brackets.

A: We corrected the reference.

R: L177 Use plural for "calibration".

A: We substituted with plural.

R: L 178 Consider to use "delivered" instead of "gave".

A: We used "delivered" instead of "gave".

R: L183 and L184 Add "all" before "cases".

A: We added "all" before "cases"

R: L186 Add "the" before "bulk ionic load".

A: We added "the".

L214 Delete the comma after ELA.

A: We removed it.

L216 Replace "snow pits samples" by "snow-pit samples".

A: I think that "snow pit" it is correct.

R: L 258-259 Delete the bracket before "Virkkunen et al." and enclose 2007 by brackets. The same for "Vega et al." (delete the bracket before and enclose the year in brackets).

A: We modified as suggested by referee.

R: L 263 Delete the bracket before "Spolaor et al." and enclose 2013 by brackets.

A: Done

R: L 278 Add "the" before "snowpack".

A: We added it

L 284 Too many brackets re-write: (0.32; Millero etal., 2008; Figure 3).

A: We modified as suggested by referee.

R: L292 Write Cl-/SO42- instead of Cl-:SO42-. Add "on" before "those".

A: Done

R: L 300-301 Delete "which is a derived variable" as this has been explained before.

A: We removed the sentence inside the commas.

R: L 312 Add "an" before "underrepresentation".

A: We added "an"

R:L 359 Add a dot after the bracket and delete "and".

A: As suggested by referee we inserted a dot and we added an "and".

R: L 368 Delete the comma after "always".

A: We removed bot commas.

R: L369 Delete the bracket before "Jacobi et al." and enclose the year by brackets.

A: Done

R:L 379 Delete the space after "this".

A: Done

R: L 400 "was" needs to be shifted behind "d".

A: We removed "d" because it is a repetition of the previous "a more detailed analysis of d by latitude".

R: L 411 Delete the comma after "higher".

A: Done

R: L 417 Replace "glaciers sites" by "glacier sites".

A: We removed "s"

R: L 428 Replace "suggests" by "suggest".

A: We removed "s"

Figure 2: Replace "glaciers zone" by "glacier zones".

A: Done

Figure 3: L 705 Delete "the" before "spring".

A: We removed "the" from now figure 5.

Figure 5: L 726 Add "the" before "oxygen".

A: We added "the" from now figure 7.

R: Table 1: Replace "Universit! de Franche-Comté" by "Université de Franche-Comté".

A: This is correct but probably something occurred during the pdf conversion.

Table 2: Replace "compare to sea water" by "compared to sea water". Please also add the unit for$\delta$18O, $\delta$D and d excess in the table captions.

A: We modified "compared to seawater" and we added the unit for the stable isotopes.

Table 4: Add a comma after Millero et al.

A: We added comma after "Millero et al."

Please also note the supplement to this comment:
https://acp.copernicus.org/preprints/acp-2020-740/acp-2020-740-AC2-supplement.pdf

---

## Author Response (AR2)

**Response to Referee 1**

R: This manuscript has been largely revised and improved based on the review comments. I have accepted the statements of authors to my review comments. I do not have any more concerning and suggestions to the manuscript. The analytical data measured in several institutions in the manuscript are high quality and reliable, because the details of analytical conditions and methods of all instruments are described. The spatial variations of the load (Table 2) and the volume-weighted mean concentrations (Table 3) of ion chemical component in the snowpack in Svalbard are quite valuable data set to research the transportation of chemical substances to Svalbard. Moreover, the phenomena of chlorine depletion and bromide enrichment are also important information to understand atmospheric conditions in Svalbard. Therefore, I believe that the quality of the manuscript is enough good for the publication as a measurement report of the ACP.

A: *Thanks for the comments. We think that your revision have improved the quality of our manuscript.*

R: Please check some parts of the manuscript before publication.Line 422. The authors used the ratio (1.17) of Cl/Na in seawater for the calculation of Cl depletion. The value of 1.17 is the mole ratio. In this manuscript, "weight units" are used for a load of chemical substances. Please confirm the calculation of the value of Cl depletion.

A: *Thanks for the comment and for this reason we clarify it adding the value (1.8 w/w) in the main manuscript. In the first paragraph of section 4.2, we used Cl-to-Na ratio as w/w while in the second part, we consider Cl depletion calculated using the formula proposed by Zhuang et al. (1999) in equivalent concentration. In the table 2, we report the value of Cl depletion, calculated using the initial concentration of Na and Cl in equivalent. I confirm our data.*

R: Please correct the valence of Na into 1 (+) in Line 113 and 523.

A: *Sorry for this terrible mistake, we corrected it.*

**Response to Referee 2**

R: The manuscript is a revised version. It is a measurement report and presents new data on the spatial variability of the snowpack chemical composition across Svalbard. The authors analysed snow pit samples taken from 7 glaciers across the archipelago for major ions and stable water isotopes aiming at improving our understanding of which processes (natural and anthropogenic) influence the snowpack chemical composition and its spatial and altitudinal distribution patterns. Comparing data on the total ionic load and stable water isotope composition among all sites the authors assessed the role of short- and long-range aerosol transport for the snowpack chemical composition on Svalbard.

The revised version has substantially improved compared to the original one and should be accepted for publication after the authors have considered some minor revisions and technical corrections.

General comments:

I thank the authors for explaining why they omitted a detailed analysis of the data on layer resolution in this manuscript. Given that the authors intend to accomplish another paper that deals with the detailed layer-by-layer investigations of the Svalbard glaciers I agree to keep this manuscript as a measurement report.

By adding information about previous studies on stable water isotopes on Svalbard to the introduction there is now a better balance between both foci of the manuscript – chemical impurities and stable water isotopes. The description of the stable water isotope analysis (chapter 2.4) and why stable water isotopes are generally used in this study has also clearly improved. I now better understand the value of including the stable water isotopes into your study. However, the main weight of the paper still lies on the chemical impurities although the title might imply something different. But since the authors now clearly state in the manuscript that the stable water isotopes are used as supporting data, I think this is not a big concern anymore.

I very much appreciate the detailed description of the different glacier sites and their characteristics as this allows to assess the presented data also with respect to the site-specific differences and/or similarities.

I thank the authors for explaining why they chose a layer-specific sampling strategy and therefore a different sampling resolution at the different sites. I now better understand the advantage of this approach with respect to the aim of the study.

*A: We would like to thanks the referees because they improved the quality of this manuscript with their revisions.*

Specific comments:

R: Title: Although the title has been modified by adding the word "seasonal" I still suggest to clearly indicate that the data refers to one single accumulation season (2015-2016), i.e. to clearly indicate the timescale of the study directly in the title. This way it would be easier for the reader to directly see what to expect from the paper.

*A: As suggested by referee, we modified the title as:*

*Measurement report: Spatial variations in ionic chemistry and water stable isotopes in the snowpack on glaciers across Svalbard during the 2015-2016 snow accumulation season*

1. Introduction:

R: L 101: I think the citation is wrong here. It should be "West et al., 2010". This reference has to be listed under "W" in the reference list, and not under "J" as it is done at the moment. Please change this.

*A: We agree with referee, we corrected it.*

R: L 105 What do you mean by "excellent environmental conditions"? Please rephrase or specify. I also think there is some information missing after "Greenland" such as "annual isotope cycles are well preserved…". Please check.

*A: We agree with referee, we had to rephrase: "The preservation of un-interrupted annual isotope cycles varies depending on the site: in sites such as central Greenland annual isotope cycles are well preserved, while in sites with high intra-seasonal variation variations or with different pre- and post-depositional processes the annual layers can be difficult to distinguish (Igarashi et al. 2001)."*

R: L 108 The subject is missing before "carried out the observation…". I guess there should be some reference added, since the next sentence starts with "These authors…". It is unclear who you mean. Please check.

*A: Thanks for the comment because we recognized a mistake probably occurred using Endnote software. We corrected the references. In the previous sentence, the correct citation was (Pohjola et al., 2002), while the missing subject was Igarashi et al. (2001).*

R: L 99-111 Has there been any previous study of stable water isotopes in surface snow from Svalbard that is as comprehensive as yours? If not, I think it is worth to stress here that there is a lack of data regarding the stable water isotope composition of surface snow and that your survey is a substantial contribution to fill that gap. I am aware that you mention the comprehensiveness und uniqueness of the survey in the following paragraph (and also at the beginning of chapter 3.2), but I think it is worth to stress also the lack of data on stable water isotopes from Svalbard snow in the introduction. This would enhance the importance of your data in this study and for future studies.

*A: Thanks for the suggestion, and we added this suggested sentences in the introduction: "At the moment, there is a lack of data regarding the stable water isotope composition of surface snow from Svalbard and this survey is a substantial contribution to fill that gap."*

2. Methods:

R: L 152-154 Rephrase to: "Even though this is the highest point in our survey, the air temperature can pass above zero during the summer resulting, although not significant, in the relocation of ions (Pohjola et al., 2002; Vega et al., 2016). "

*A: Thanks, we modified it as suggested.*

R: L 160-164 I would very much appreciate if you could add the information provided in your answers that "all snow pits have been collected from the glacier central line in order to minimise the side accumulation effect due to orography" as this is a valuable detail regarding the sampling strategy.

*A: As suggested by referee, we added the sentence reported in our answers.*

3. Results:

L 274-275 I am not sure whether I caught right what the authors want to say here as the phrasing is a bit confusing. Please rephrase to: "… since deposition can still occur before the beginning of the snow melt season."

*A: Yes, the meaning of sentence is correct and we modified it as suggested.*

R: L 278 What do you mean with "the other regions"? Please specify or delete "the" to make it unspecified.

*A: We removed "the" to make it unspecified.*

R: L 307 Stable water isotopes are shown in Table 3, not in Table 2 as indicated in the text. Please correct this.

*A: Thanks, this a mistake come from the previous version of Tables. We corrected it.*

R: L 316-317 For clarity please add "of all snow pits" after "mean $\delta$18O and $\delta$D values", as it was in the previous version of the manuscript.

*A: We add "of all snow pits", as suggested by referee.*

Actually, for AF I do not entirely agree that there is no statistical difference between the mean $\delta$18O and $\delta$D values of the different snow pits as a 2‰ difference in $\delta$18O and 15‰ in $\delta$D is quite significant to me. For example, from Table 3 I can see that at ALB the differences between the different snow pits are even lower than at AF, although the altitudinal distances between the different snow pit sampling sites is comparable to those of AF. Please justify your statement. Otherwise, I think it is worth to mention that the altitudinal pattern in the stable water isotope composition of the snowpack at AF is different to the other glaciers as it was done in the previous version of the manuscript. I then suggest to add just one sentence about possible reasons for the different pattern at AF to the discussion (chapter 4.4, see comment below).

*A: We agree with referee and we add a sentence to suggest a possible reason.*

*"The relationship with elevation was is similar for both isotopic ratios in the collected dataset, with except AF that the isotopic signals might be influenced by additional processes since it is an isolated ice cap mainly surrounding from ocean or sea ice in winter."*

4. Discussion

L 454 I did not find the data from the Norwegian Meteorological Institute in the reference list. Please cite properly, e.g., cite a related publication or the webpage and last access date of the data.

A: To clarify the reference, we added the web site "(https://cryo.met.no/en/sea-ice)."

R: L 462-463 Please make a new sentence for better readability: "… on Spitsbergen or on Austfonna. The relationship with elevation is …".

*A: We did the suggested split.*

R: As mentioned above, please add here that the relationship with elevation, i.e., that mean δ18O and δD values decrease with altitude, is true for all glaciers except AF and discuss shortly some possible explanations, that might be also speculative, for this.

A: As mentioned above, we added this sentence: "The relationship with elevation was is similar for both isotopic ratios in the collected dataset, with except AF that the isotopic signals might be influenced by additional processes since it is an isolated ice cap mainly surrounding from ocean or sea ice in winter."

R: L 459 Same as before. Stable water isotopes are shown in Table 3 instead of Table 2.

A: Sorry, we corrected it.

R: L 486 Accumulation zones of which glaciers? All seven glaciers? Please specify.

A: We specified "all seven glaciers"

5. Conclusions

I strongly suggest to write the paragraph from L 524 to L 535 in present tense.

A: We modified the paragraph in the present tense.

Technical corrections:

R: Please make also sure to use present tense when discussing your results. Change to present tense in the following sections/sentences: L 343-346, L 350, L 371-373, L 381-385, L394, L 411-413, L 431-434, L 445, L 457-459, L 462-463, L 468-471, L 482-483, L 488-492, L 498-502, L 505-507

A: As suggested by referee, we use the present tense in all suggested sections.

In some paragraphs the reference to the Table with the data you are discussing is missing. For clarity, please add the reference to Table 5 in the following parts: sentence in L 358-359, sentence in L 382-385, sentence in L 411-413; pleas add the reference to Table 3 in L 467-469.

A: As suggested by referee, we added the reference to the Tables in the suggested points.

L 86: Rephrase "summer melting" to "summer melt".

L 102 Change "snowpits" to "snowpit" (Singular).

L 103 Change "depth/time relation" to "depth-time relationship".

L 106 Change "variation" to "variations" (Plural).

L 109 Add space between "could" and "not".

L 116 Add "the" before "snowpack".

L 118 Add a verb – "is" or "was" – before "… the most comprehensive survey…".

L 134 Add space between "ice" and "cap".

L 135 A verb is missing before "… one main central dome…". Probably "has" needs to be added.

L 142 Add space between "400" and "m".

L 144 Change "thinness" to "thickness". Rephrase to "… and a maximum of 450 m". Add dot after the brackets.

L 150 Add "the" before "island".

L 151 Finish the sentence after "500 m" and start a new sentence. There are also some letters mixed up. Please correct to: "The total accumulation area of the entire …". Change "in the beginning" to "at the beginning".

L 172 Please rephrase to: "… and were directly filled into …".

L 176 Please rephrase to: "… when we compare three different areas of the same glacier…".

L 223-224 Change to "… are reported by Barbaro et al. (2017)."

L 227 Delete "and" before "coupled to a …"

L 246 Change "to evaluate" to "to be evaluated".

L 259 Change "determinations" to "determination" (Singular)

L 274-275 Please rephrase to: "…can still occur before snow melt begins."

L 309 Add "the" before "SWE of each layer…"

L 315 Rephrase to: On KVG, ALB, HDF and LF, $\delta18O$ and $\delta2H$ values in snow…".

L 317 Add "on" before "AF, WB and HB there was …".

L 324-325 Change to "Virkkunen et al. (2007) and Vega et al. (2015a) quantified…".

L 329 Delete the space between "is" and "wide".

L 352 Change to "(0.32; Figure 5; Millero et al., 2008)".

L 383 Change "correlations between the bulk loads…" to "correlations of the bulk loads …"

L 407 Add "the" before "snowpack".

L 413-414 There is a doubling of the reference. Change to "Whitlow et al. (1992) found an …".

L 416 Change "as well" to "as well as".

L 426-427 Rephrase to: "reaching more northerly locations."

L 440 Change to "findings of Jacobi et al. (2019) ….".

L 471 Rephrase to "… shows that significantly different values are only obtained in snow pits …".

L 472 Delete one dot after "snow pots".

L 502 Replace "suggest" by "indicate".

L 503 Change "warm event" to "warm air event".

L 504 Rephrase to: "On the contrary, when cold air masses (Arctic type) dominate, snowfall events are relatively limited due to the low air humidity causing a lower efficiency of wet scavenging."

L 507-508 Delete "is" before "should".

L 509 For better readability start a new sentence after "wet deposition" instead of separating sentences by semicolon.

L 510 Change to "mask the proposed relationship".

L 518 Add "a" before correlation.

L 539-542 Please rephrase to: "These findings confirm that the optimal sites to study the effects of long-range pollution deposition in Svalbard are those at higher elevations, such as the accumulation zones of HDF or LF, because they are the sites least impacted by local aerosol emissions."

L 544 Delete "there" after "chemical composition".

L 794 (Figure 1) Rephrase to: "… except on KVG glacier where an extra snowpit was sampled within the ablation zone."

L 814 (Figure 3) Add a dot at the end of the Figure captions.

Figure 7 In the Figure change "significativity" to "significance".

*A: Thanks for the precise comments because they help us to improve the quality of our manuscript. We modified the manuscript, following the referee suggestions about the technical corrections.*

L 307 Change to "mean $\delta18O$ and $\delta2H$ values are …"

*A: Sorry but we can not do that because the meaning is different. Here, we have to define the meaning of "the SWE-weighted mean $\delta18O$ and $\delta2H$".*

R: L 472-475 It is not clear where the bracket at the end of the sentence starts. Please check which information should be in brackets. I also suggest to make two sentences instead of separating the different parts by semicolon.

*A: As suggested by referee, we divide the sentences in two parts as follows: "This is confirmed by the Kruskal-Wallis test, i.e. rank-based ANOVA, calculated with two groups of d values divided by the latitude threshold 79.2°N ($z = 4.23$, $p < 0.04$). In fact, drawing the latitude threshold anywhere between 78.7 and 79.7 °N, a statistically significant difference with $p < 0.05$ is obtained."*

R: L 894 (Table 5) From the phrase "estimated based from" it is not clear to me what you want to say: "Non-sea-salt (nss) components were estimated based on seawater ratios to …" or "Non-sea-salt (nss) components were estimated from seawater ratios to …"? Please check.

*A: Thanks for the comment. To clarify the sentence, we modified as follows: "Non-sea-salt (nss) components were estimated based from seawater ratios: $Ca2+/Na+$ is 0.038 while and $SO42-/Na+$ is 0.252 (w/w; Millero et al., 2008).*